# Species196: A One-Million Semi-supervised Dataset for Fine-grained Species Recognition

**Wei He, Kai Han**\*, **Ying Nie, Chengcheng Wang, Yunhe Wang**\*
Huawei Noah's Ark Lab
{hewei142,kai.han,yunhe.wang}@huawei.com

## Abstract

The development of foundation vision models has pushed the general visual recognition to a high level, but cannot well address the fine-grained recognition in specialized domain such as invasive species classification. Identifying and managing invasive species has strong social and ecological value. Currently, most invasive species datasets are limited in scale and cover a narrow range of species, which restricts the development of deep-learning based invasion biometrics systems. To fill the gap of this area, we introduced Species196, a large-scale semi-supervised dataset of 196-category invasive species. It collects over 19K images with expert-level accurate annotations (*Species196-L*), and 1.2M unlabeled images of invasive species (*Species196-U*). The dataset provides four experimental settings for benchmarking the existing models and algorithms, namely, supervised learning, semi-supervised learning, self-supervised pretraining and zero-shot inference ability of large multimodal models. To facilitate future research on these four learning paradigms, we conduct an empirical study of the representative methods on the introduced dataset. The dataset is publicly available at `https://species-dataset.github.io/`.

## 1 Introduction

Invasion biometrics play a critical role in the identification and management of invasive species, which are non-native organisms capable of causing detrimental impacts on the environment, economy, and human health [18]. Traditionally, invasive species recognition systems have relied on trained experts who analyze an animal or plant's physical characteristics, or use DNA testing [11]. However, these conventional methods require specialized equipment, expert knowledge, and are often time-consuming and costly, while also posing potential risks to creature well-being [22]. Recent methods [53, 91, 2], utilizing computer vision techniques and deep learning algorithms, provide a cost-effective and efficient alternative solution for invasive species identification.

As computer vision-based methods continue to gain traction, the need for high-quality biometrics data becomes increasingly crucial for effective invasive species recognition. In contrast to traditional visual tasks such as image classification or object detection, which benefit from large-scale datasets like ImageNet [12], Open Images Dataset[48], COCO[55], and Objects365 [72], acquiring diverse, high-quality datasets for training and evaluating invasive species recognition algorithms remains a challenge.

In response to this challenge, we introduce *Species196-L*, a new fine-grained hierarchical dataset focusing on regional invasive species. The "L" donates that the dataset is fine labeled. *Species196-L* collects 19236 images for 196 invasive species listed in the *Catalogue of Quarantine Pests for Import Plants to China* [65]. In addition, We also provide bounding box annotation and detailed

---

\*Corresponding Author

37th Conference on Neural Information Processing Systems (NeurIPS 2023) Track on Datasets and Benchmarks.

multi-grained taxonomy information for each image, which assists users in building more effective invasive species biometrics systems.

Self-supervised and semi-supervised learning are particularly useful when labeled data is costly or difficult to collect. To further enhance our *Species196* dataset, we utilize the publicly available LAION-5B dataset [71] to create a new, large-scale 1.2M-image subset based on *Species196-L*'s invasive species, named *Species196-U*. The "U" signifies that the dataset is unlabeled. In *Species196-U*, each image is retrieved from the LAION-5B dataset [71] using a pre-trained OpenCLIP $_{ViT-L/14}$ [57] image encoder, based on the original finely annotated *Species196-L*. This dataset expands the amount of data available for invasive species research and provides a valuable domain-specific resource for unsupervised, self-supervised, or multi-modal learning.

Our main contributions are summarized as follows:

• To the best of our knowledge, we have constructed the largest-scale invasive species biometrics dataset (*Species196-L*), covering a wide range of 196 species. This dataset not only encompasses different life cycles but also provides detailed taxonomic information for each species, ranging from Kingdom and Phylum to Genus and Species.
• We present a large-scale, domain-specific unlabeled dataset, *Species196-U*, which is highly related to images from *Species196-L*, containing 1.2 million image-text pairs. This dataset serves as an ideal testbed for evaluating various pretraining and multi-modal methods.
• We perform extensive experiments on our proposed datasets using a variety of methods, including state-of-the-art CNN and transformer networks, specifically designed fine-grained visual recognition techniques, self-supervised and semi-supervised approaches, and also CLIP and multi-modal large models' zero-shot inference performance on *Species-L*. These comprehensive experiments establish a benchmark for Species196.

## 2 Related work

### 2.1 Invasion biometrics

Invasive species cause ecological change, harm biodiversity [8], and present significant threats to global agriculture [67] and economic welfare [46]. Invasive biometrics plays a vital role in identifying and managing these species. Conventional invasive species identification methods (*e.g.* physical analysis and DNA testing) are costly, time-consuming, and heavily relied on expert knowledge and experience[11, 22].

In recent years, there has been a growing interest in using computer vision (CV) for invasive species recognition, as it offers more efficient, cost-effective, and scalable alternatives to traditional methods. These methods can be divided into two types - handcrafted and deep feature based. Handcrafted based methods adopting feature extractors like SIFT [61] and HOG[10]. Li *et al.* [53] propesed a HOG and SVM based methods for identifying invasive Asian Hornets [13], which extracting image features using the HOG algorithm, and using the SVM algorithm for target detection. More recently, deep learning techniques have shown its potential in this field. Chowdhury *et al.* [7] suggested an approach for identifying aquatic invasive species by utilizing an autoencoder feature extractor, coupled with a classifier trained to distinguish between invasive and non-invasive species. Deep convolutional neural netwok is also used by Ashqar *et al.* [2] to identifying images of invasive hydrangea, and by Huang *et al.* [40] to develop a fast and accurate detection technology to identify invasive weeds.

Although deep learning-based methods are becoming increasingly popular, they heavily rely on high-quality annotated data compared to handcrafted-based methods, especially in practical applications where high-quality labeled data is often the performance bottleneck for invasion biometrics or other pest recognition applications.

### 2.2 Related datasets

In this section, we provide an overview comparing our proposed Species196 dataset with other related species recognition and fine-grained datasets. The comparison overview is shown in Table 1.

Pink-Eggs [89] comprises images accompanied by bounding box annotations of the invasive species Pomacea canaliculata, Cional17 [20] is a semantic segmentation dataset providing pixel-level annotations concerning invasive species in marine environments. However, these two invasive biometrics

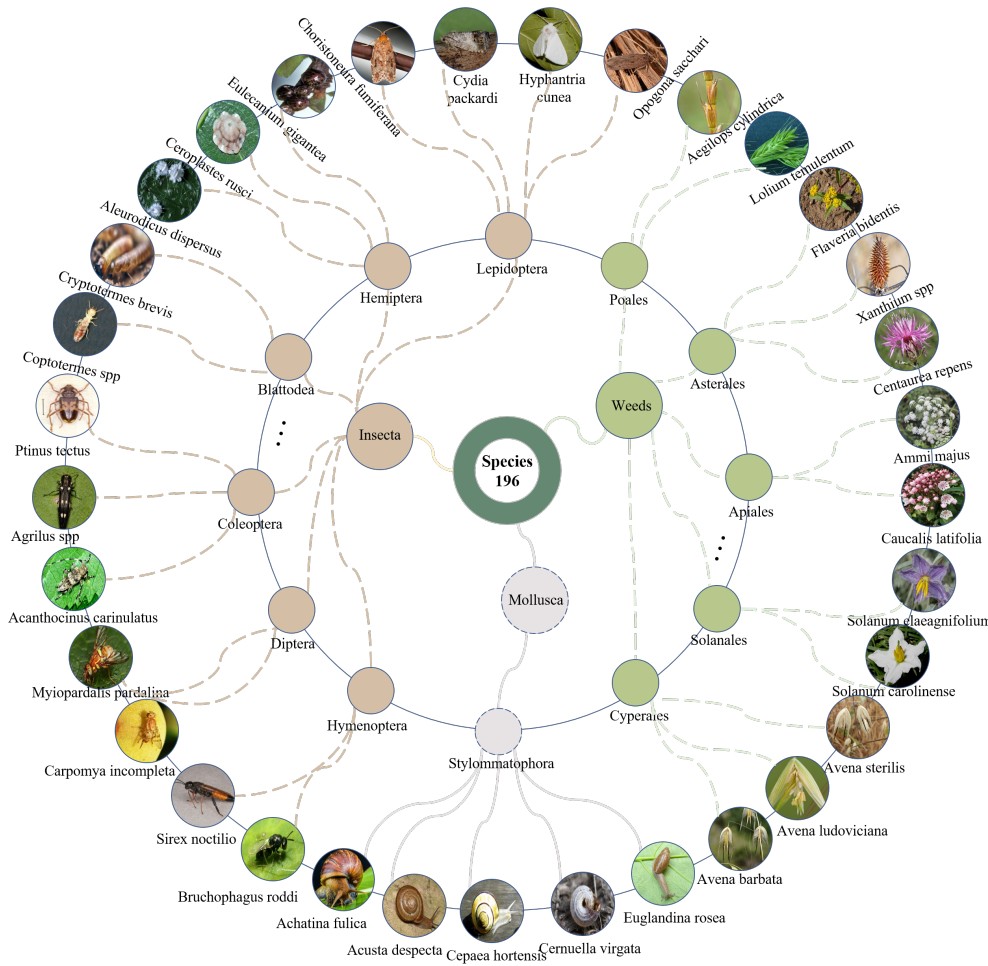

Figure 1: Taxonomy of a portion of the *Species196-L* dataset's taxonomy hierarchical system.

datasets possess a rather limited number of samples and focus only on specific types of organisms. In comparison, our *Species196-L* dataset offers nearly 20,000 samples, covering 196 diverse invasive species, making it particularly valuable in various real-world applications such as customs and border crossings quarantine.

There are also some datasets that share the same super-categories, such as weeds and insects, with *Species196-L*. Some of them [86, 24, 66] have fewer sample numbers and species compared to *Species196-L*, while other datasets [60, 1, 36] are not publicly available so far. IP102 [85] and CWD30 [43] are two recent representative datasets in the field of insect and crop-weed. IP 102 covers 102 species of common crop insect pests with over 75, 000 images, captures various growth stages of insect pest species. CWD30 comprises over 219,770 high-resolution images of 20 weed species and 10 crop species, encompassing various growth stages, multiple viewing angles, and build a hierarchical taxonomic system foe these weeds and crops. Following the best practices of IP102 and CWD30, our *Species196-L* dataset covers various life stages of insects and establishes a comprehensive taxonomy system, from domain, kingdom, down to order, family, genus, and species, aiding in the creation of a precise and robust recognition system.

As a challenging benchmark dataset for fine-grained image recognition, we also compare *Species196-L* to other popular datasets in this field. Oxford Flowers [64], CUB200 [35], FGVC Aircraft [62] and Stanford Cars [47] are popular fine-grained image classification datasets with different categories of flowers, birds and cars, respectively. However, these datasets do not involve the challenge of distinguishing species at different life stages and do not provide multi-grained taxonomy information.

Table 1: Comparison of related species recognition and fine-grained datasets

| Dataset | Year | Meta-classes | Categories | Samples | Multiple life-cycle | Taxonomy |
|---------|------|-------------|-----------|---------|--------------------|---------|
| Pink-Eggs [89] | 2023 | Mollusca | 3 | 1261 | N | N |
| Ciona17 [20] | 2017 | Mollusca, Chordata | 5 | 1472 | N | N |
| IP102[3 [85] | 2019 | Insects | 102 | 75,222 | Y | N |
| Pest ID [60] | 2016 | Insects | 12 | 5,136 | N | N |
| Xie *et al.* [86] | 2018 | Insects | 40 | 4,500 | N | N |
| Alfarisy *et al.* [1] | 2018 | Insects | 13 | 4,511 | N | N |
| Deep Weeds [66] | 2019 | Weeds | 9 | 17,509 | N | N |
| Plant Seedling [24] | 2017 | Weeds | 12 | 5,539 | N | N |
| CNU [36] | 2019 | Weeds | 21 | 208,477 | N | N |
| CWD30 [43] | 2023 | Crops, Weeds | 30 | 219,778 | Y | Y |
| Oxford Flowers [64] | 2008 | Plants | 102 | 8,189 | N | N |
| CUB200 [35] | 2011 | Birds | 200 | 11,788 | | N |
| Stanford Cars [47] | 2013 | Cars | 196 | 19,184 | | N |
| FGVC Aircraft [62] | 2013 | Aircrafts | 100 | 10,000 | | N |
| iNat2017 [37] | 2017 | Plants, Animals | 5,089 | 859,000 | Y | Y |
| iNat2021 [38] | 2021 | Plants, Animals | 10,000 | 3,286,843 | Y | Y |
| **Species196** | 2023 | Mollusca, Weeds, Insects | 196 | 19,256 + 1,200,000 (unlabeled) | Y | Y |

The iNat2017 dataset [37] introduced a large-scale, fine-grained image classification and detection dataset with 859K images from 5,089 species, emphasizing the importance of few-shot learning. The iNat2021 dataset [38] expanded this to 2.7M images from 10K species and highlighted the advantages of self-supervised learning methods for fine-grained classification. In contrast to the iNat2017 and iNat2021 datasets, which are derived from user-contributed observations on the iNaturalist community[44], the scale and collection method of our Species196 dataset align more closely with real-world applications. Moreover, Species196 provides a challenging testbed for the research community to explore how to improve model's performance through limited meticulously labeled data and large-scale unlabeled data.

## 3 Dataset

### 3.1 Taxonomic system establishment

Figure 1 illustrates a portion of the hierarchical taxonomy system for *Species196-L*. All species included in our collection are sourced from the *Catalogue of Quarantine Pests for Import Plants to China* [65]. With the aim to help construct an effective and efficient computer vision-based invasion biometrics system, we selected insects, weeds, and mollusks that are visually observable and also easily collectable using mobile devices.

We have provided a taxonomy system that includes comprehensive hierarchical classification information within the dataset. This information spans from the domain, kingdom, and phylum levels down to the order, family, genus, and species of each invasive species. We hope that the inclusion of biological taxonomy data, incorporating prior knowledge in this field, will enable users of the Species196 dataset to explore and construct more efficient, effective, and robust invasive species identification systems.

### 3.2 Image collection, filtering, split, and annotation of *Species196-L*

In this work, we utilize public available information on the internet as the source for collecting images of invasive organisms. In addition to directly using search engines, we also place a strong emphasis on exploring various global biological image repositories for image collection. Examples of the image repositories we accessed include the iNaturalist community [44], the Global Invasive Species Database (GISD) [23], BugGuide [5], Biocontrole [4], and more. These diverse and trustworthy sources provided a comprehensive and rich set of images for the *Species196-L*.

Our team of five members utilized taxonomy information to search for images using both common names and scientific names. We collected images of insects at various stages of their life cycle,

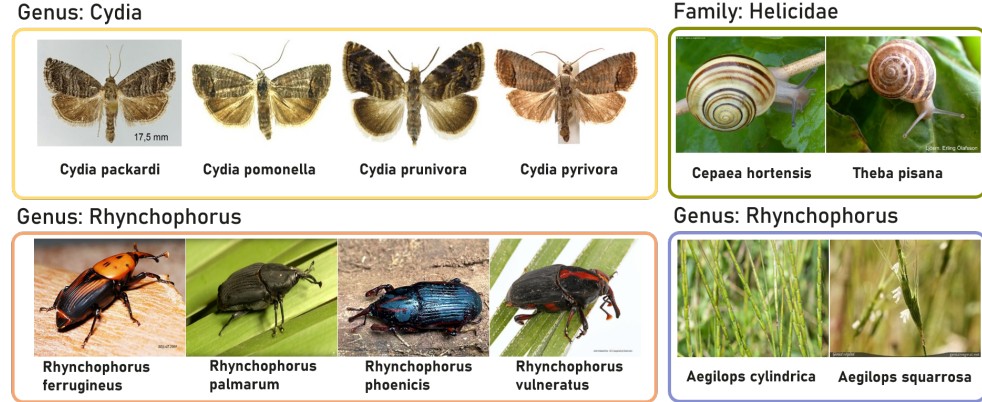

Figure 2: Display images of species belonging to the same genus or family.

such as eggs, larva, pupa, and adult (see Figure 3), because each stage can cause negative impacts on ecosystems and the economy. We also removed images that contained more than one invasive species category. Our dataset is a multi-grained dataset with class granularity based largely on both species and genus, and when processing categories primarily by genus, we have balanced the number of categories within each genus as much as possible to help ensure a relatively even distribution. Subsequently, we removed images with a size lower than 128x128 and those containing private information such as human faces. We collected 19,236 images for 196 invasive species. Following the approach of Stanford Cars [47] and CUB200 [35], we evenly divided our train and test sets at a 1:1 ratio, resulting in a train size of 9,676 images and a test size of 9,580 images.

In real-world pest control, precise identification and location of pests are vital. This task, often complicated by cluttered backgrounds and multiple pests in one image, aids in effective pest management. After collection and filtering, we labeled 19,236 images with 24,476 bounding boxes in the COCO format [54] for *Species196-L*, which took five individuals about six months.

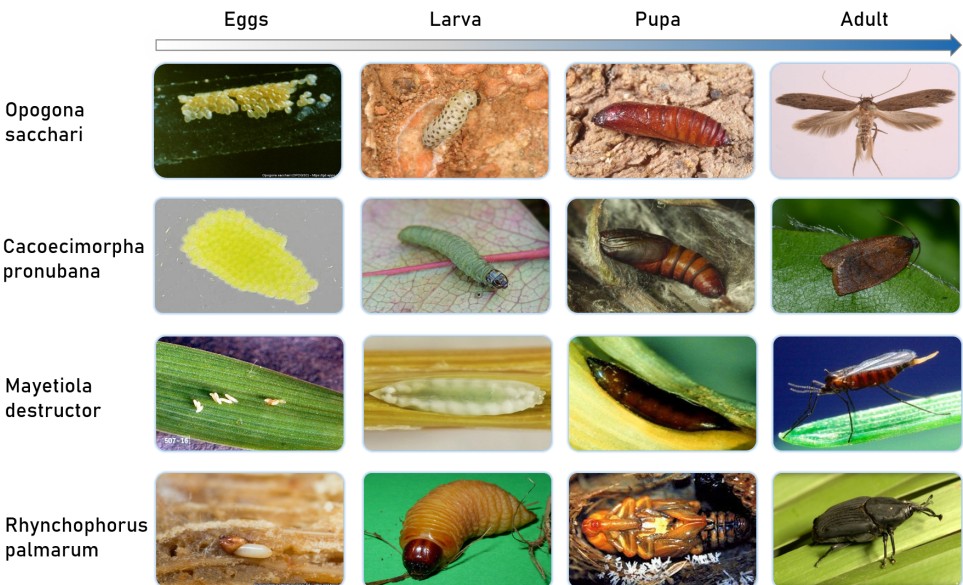

Figure 3: Invasive insect life cycles: egg, larva, pupa and adult.

### 3.3 Challenges of fine granularity and data imbalance

*Species196-L* share common challenges like other fine-grained datasets such as high similarity between different classes and low variability within each class. Figure 2 shows example of classes

belonging to the same genus, which are difficult to distinguish. Data imbalance is another challenge. As illustrated in Figure 5, although our dataset contains species from 21 different orders, the majority of them are concentrated in a few orders, such as Coleoptera, Lepidoptera, and Hemiptera.

## 3.4    Data collection of *Species196-U*

Prior research has demonstrated that domain-adaptive pretraining has the potential to improve performance in both the natural language [38] and computer vision [27] fields. In recent years, the introduction of the CLIP (Contrastive Language-Image Pre-training) method [68] and the emergence of web-harvested image-text data like LAION5B [71] have provided us with the opportunity to build large-scale domain-specific data at minimal cost.

In this work, we utilize the clip-retrieval [3] to retrieve the large-scale *Species-U* dataset from LAION-5B. We first evaluated various retrieval methods and discovered that image-to-image retrieval surpassed other text-to-image retrieval approaches in terms of relevance. These approaches included retrieval based on common names, scientific names, and artificial descriptions. Specifically, we randomly sampled three images for each category and retrieved 8,000 unlabeled images per class. For insects with different life cycles, we additionally sampled eggs, larvae, and pupae once for each species. After removing duplicates, we obtained a final dataset consisting of approximately 1.5 million images.

For comparison with popular experiments using ImageNet-1K, we chose 1.2 million image-text pairs from LAION-5B, a number smaller than that in ImageNet-1K.

## 3.5    Retrieved Example Images from the  *Species196-U* Dataset

We use image-retrieval for creating  *Species196-U*. For each category, we randomly sampled three images and retrieved 8,000 unlabeled images per class from LAION5B. As shown in Figure 4, even at the $5,000_{th}$ image sorted by descending similarity scores, the retrieved image remains highly relevant to the original image.

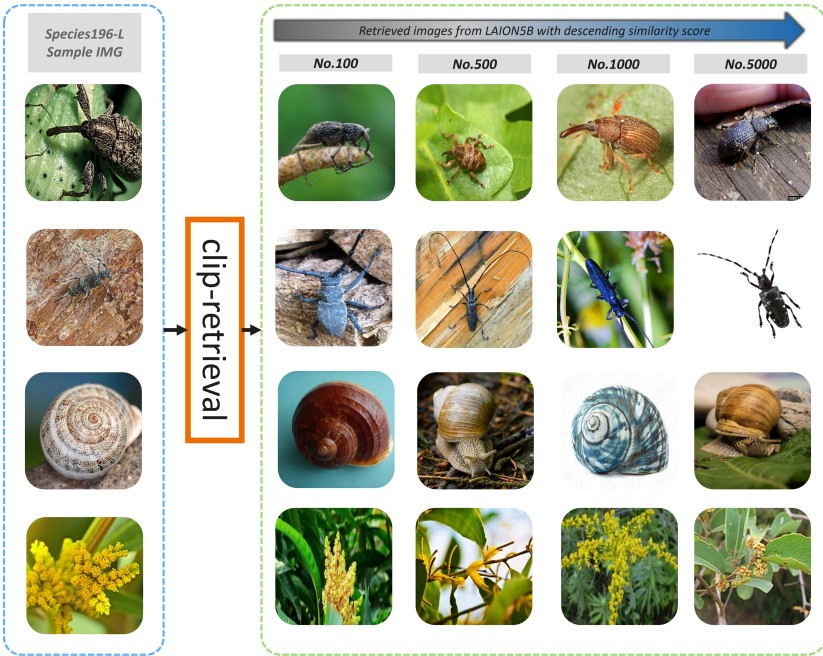

Figure 4: Clip-retrieval process of  *Species16-U* from  *Species196-L*. Displaying similarity scores in descending order, we show items No. 100, 500, 1000, and 5000.

# 4 Experiment

To benchmark *Species196-L*, we assessed methods with CNN, Transformer, or hybrid backbones of varying scales, including fine-grained models. We also experimented with unsupervised and semi-supervised methods on the *Species196-U* dataset. Moreover, we examined zero-shot inference capabilities by testing the performance of CLIP and several large multi-modal models.

## 4.1 Experiment on supervised learning methods

We first tested mainstream visual backbones and several recent fine-grained classification methods. Keeping the potential deployment of invasion biometrics systems on the edge side in mind, we specifically compared lightweight models. Detailed comparison results are presented in Table 2. As for experimental details, we utilized timm [83] as the codebase to evaluate different models. For networks with the same architecture and similar complexity (*e.g.*, CNN-based, Transformer-based, and Hybrid structures), all models were trained for 300 epochs with the same input resolution of 224×224 for fair comparison. For fine-grained classification network methods, we selected MetaFormer-2 [14], TransFG [31], and IELT [90], which have demonstrated promising performance on other fine-grained classification datasets CUB200 [35].

Our experimental results indicate that ResNet50 [33] performs best when trained from scratch. Furthermore, on the *Species196-L* dataset, pre-training and transfer learning significantly outperformed training from scratch for models of all sizes, particularly transformer-based and hybrid networks such as TNT-S [30] and CMT-S [26]. Among the top-performing models in small, medium, and large-scale comparisons are MobileViT-XS [63], MaxViT-T [79], and MViTv2-B [52]. MetaFormer-2 [14] achieved impressive accuracies of 87.69% withand 88.69% on the *Species196-L* dataset, respectively. This performance was achieved through pre-training with ImageNet-1K and Imagenet-22K. MetaFormer-2 [14] achieved impressive performance on our Species196-L dataset, reaching state-of-the-art (SOTA) accuracy of 88.69% with a resolution of 384×384 input size. It also achieved a top-1 accuracy of 87.69% in the 224×224 resolution.

We also conducted experiments on popular object detection networks, ranging from CNN-based methods such as Faster-RCNN [70] and YOLOX-L [21], to DETR [6]-like Deformable DETR [97] and DINO [94]. Using Imagenet-1k pre-trained weights, experiment results show that DINO with Swin-Base backbone achieved the best accuracy, while Deformable DETR with ResNet50 backbone attained a balance between accuracy and parameters. The comparison detail is shown in Table 3.

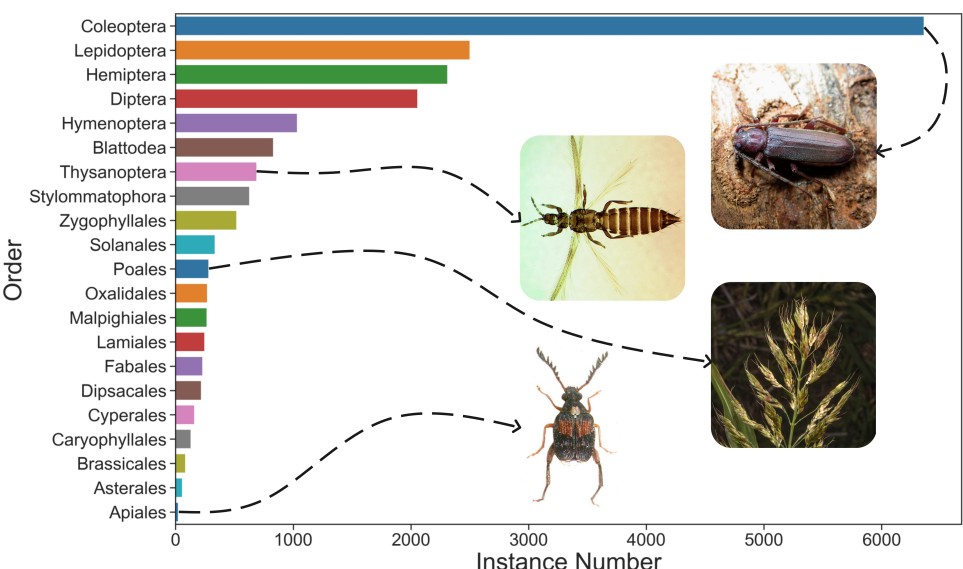

Figure 5: Imbalance distribution of *Species196-L* based with order-level granularity.

Table 2: Comparison of different modern backbones and fine-grained methods. [†] and [‡] donates using Imagenet-1K, Imagene22K pretrained weight, separately. The rest are trained from scratch.

| Model | Resolution | # Params. | # FLOPs. | Top - 1 ACC. | Top - 5 ACC. | $F1_{MACRO}$ |
|---|---|---|---|---|---|---|
| MobileViT-XS [63] | $224^2$ | 2.3 M | 0.7 G | 64.11 / 78.55[†] | 83.51 / 91.92[†] | 53.52 / 69.01[†] |
| GhostNet 1.0 [29] | $224^2$ | 5.2 M | 0.1 G | 62.75 / 76.02[†] | 82.58 / 90.77[†] | 51.30 / 64.93[†] |
| EfficientNet-B0 [75] | $224^2$ | 5.3 M | 0.4 G | 62.88 / 78.26[†] | 81.66 / 91.60[†] | 53.13 / 66.91[†] |
| MobileNetV3 Large 1.0 [39] | $224^2$ | 5.4 M | 0.2 G | 62.75 / 77.83[†] | 81.46 / 90.77[†] | 49.99 / 66.50[†] |
| RegNetY-4GF [69] | $224^2$ | 20.6 M | 4.0 G | 43.01 / 82.25[†] | 69.02 / 93.71[†] | 28.99 / 71.24[†] |
| Deit-S [78] | $224^2$ | 22 M | 4.6 G | 36.89 / 77.21[†] | 56.79 / 91.52[†] | 29.35 / 65.25[†] |
| TNT-S [30] | $224^2$ | 23.8 M | 5.2 G | 38.66 / 80.67[†] | 59.14 / 93.17[†] | 30.67 / 69.34[†] |
| CMT-S [26] | $224^2$ | 25.1 M | 4 G | 40.86 / 81.12[†] | 60.10 / 93.32[†] | 33.25 / 70.40[†] |
| Resnet50 [33] | $224^2$ | 25.6 M | 4.1 G | 64.32 / 78.11[†] | 81.70 / 91.91[†] | 53.31 / 67.29[†] |
| Swin-T [58] | $224^2$ | 28 M | 4.5 G | 46.88 / 81.66[†] | 68.57 / 93.52[†] | 37.30 / 71.20[†] |
| Convnext-T [59] | $224^2$ | 29 M | 4.5 G | 46.36 / 78.94[†] | 68.59 / 92.44[†] | 37.16 / 70.43[†] |
| MaxViT-T [79] | $224^2$ | 31 M | 5.6 G | 52.19 / 83.35[†] | 72.12 / 94.16[†] | 42.40 / 62.56[†] |
| MViTv2-B [52] | $224^2$ | 52 M | 10.2 G | 46.22 / 83.79[†] | 66.21 / 94.81[†] | 35.83 / 72.94[†] |
| Resnet200-D [34] | $224^2$ | 65 M | 26 G | 51.35 / 82.11[†] | 73.07 / 94.76[†] | 37.70 / 70.61[†] |
| VIT-B/32 [16] | $224^2$ | 86 M | 8.6 G | 32.59 / 74.68[†] | 53.76 / 89.76[†] | 25.20 / 63.38[†] |
| Swin-B [58] | $224^2$ | 88 M | 15.4 G | 48.72 / 82.88[†] | 69.71 / 94.30[†] | 39.28 / 72.04[†] |
| Pyramid ViG-B [28] | $224^2$ | 82.6 M | 16.8 G | 61.59 / 82.82[†] | 82.93 / 91.96[†] | 34.27 / 72.40[†] |
| MetaFormer-2 [14] | $224^2$ | 81 M | – | 87.69[‡] | – | |
| MetaFormer-2 [14] | $384^2$ | 81 M | – | 88.69[‡] | – | |
| TransFG [31] | $224^2$ | 85.8 M | – | 84.42[‡] | – | |
| IELT [90] | $448^2$ | 93.5 M | – | 81.92[‡] | – | |

## 4.2 Experiment on semi-supervised and self-supervised learning methods

### 4.2.1 Semi-supervised learning

The Noisy Student approach [87] leverages a teacher model to generate pseudo labels for unlabeled data, training a larger student model with both labeled and unlabeled sets. We used a similar semi-supervised method on our smaller-scale *Species196-U* dataset compared to YFCC100M [76] and JFT [74]. Following this approach, we incorporated data augmentation and dropout for noise injection in training, using a model of equal or smaller size for labeling unlabeled data, then training a larger model. Table 4 shows that using Noisy Student training on our 1.2 million unlabeled data enhances performance when the student model is larger. However, if the student network is identical to the teacher's, accuracy may decline.

### 4.2.2 Unsupervised learning

In recent years, with methods like MAE [32] , SimMIM [88] and LocalMIM [81], masked image modeling unsupervised learning has received increasing attention, ConvNeXt V2 [84] and SparK [77] further successfully bring MIM into CNN networks. Inspired by some success works in the NLP domain (*e.g.* [27], [51]), we constructed the *Species-U* dataset for the Invasion Biometrics task and conduct an empirical study on different MIM methods(see Table 5). We found that even with less data, pretraining on our *Species-U* dataset can surpass the Imagenet-1K pretrained models on both CNN-based and Transformer-based networks with the same or fewer pretraining epochs. The results

Table 3: Comparison of average precision performance of object detection methods. IoU threshold range of 0.5 to 0.95.

| Methods | Backbone | #Params. | AP | $AP_S$ | $AP_M$ | $AP_L$ |
|---|---|---|---|---|---|---|
| Faster-RCNN [70] | Resnet50 [33] | 40 M | 44.7 | 5.2 | 24.9 | 46.0 |
| YOLOX-L [21] | CSPNet [80] | 54 M | 50.3 | 9.9 | 31.0 | 51.3 |
| Deformable DETR [97] | Resnet50 [33] | 40 M | 56.9 | 10.8 | 36.9 | 58.3 |
| DINO [94] | Resnet50 [33] | 47 M | 57.7 | 9.1 | 40.0 | 59.2 |
| DINO [94] | Swin-Base [58] | 109 M | 67.7 | 19.1 | 46.5 | 69.6 |

Table 4: Comparison using a student model with the same size or with a larger size. Student Acc. represents the top-1 accuracy of the student network at the end of the last iteration.

| Teacher | Teacher Acc. | Student | Student Acc. |
|---------|--------------|---------|--------------|
| ResNet18 | 71.1 | ResNet18 | 70.6 |
| ResNet18 | 71.1 | ResNet34 | **73.4** |
| ResNet34 | 72.3 | ResNet34 | 72.0 |

Table 5: Comparison of different model architecture with different MIM methods. PT/FT donates pre-train and fine-tune stage.

| Backbone | PT method | PT data | PT epoch | FT epoch | Top - 1 ACC. | Top - 5 ACC. |
|----------|-----------|---------|----------|----------|--------------|--------------|
| ***Vision Transformer Backbone*** | | | | | | |
| VIT-B | SimMIM [88] | ImageNet-1K | 800 | 100 | 80.9 | **94.9** |
| | SimMIM [88] | *Species-U* | 800 | 100 | **81.0** | 94.6 |
| Swin-B | SimMIM [88] | ImageNet-1K | 800 | 100 | 80.5 | 94.5 |
| | SimMIM [88] | *Species-U* | 800 | 100 | **81.6** | **95.0** |
| ***Convolutional Backbone*** | | | | | | |
| Resnet50 | SparK [77] | ImageNet-1K | 1600 | 300 | 73.2 | 88.6 |
| | SparK [77] | *Species-U* | 800 | 300 | 73.2 | **88.9** |
| ConvNeXt V2-T | FCMAE [84] | ImageNet-1K | 1600 | 300 | 77.1 | 92.6 |
| | FCMAE [84] | *Species-U* | 1600 | 300 | **79.3** | **93.1** |
| ConvNeXt V2-B | FCMAE [84] | ImageNet-1K | 1600 | 300 | 79.9 | 93.1 |
| | FCMAE [84] | *Species-U* | 800 | 300 | **81.2** | **94.0** |

also indicate that models with stronger hierarchical structures, such as traditional convolutional networks and Swin Transformers, tend to benefit more from pretraining on the *Species-U* dataset.

## 4.3 Experiment on multimodal large language models

Recently, there has been a surge of interest in the field of Multimodal Large Language Models (MLLM) [93, 82, 82, 45, 95]. These models leverage the ability of Large Language Models (LLMs) to perform a variety of multimodal tasks effectively. However, we discovered that there are few existing benchmarks that can assess the MLLM's ability to handle fine-grained knowledge, let alone test its performance across various levels of granularity.

We designed question-and-answer tasks using images from *Species196-L*, which included both multiple-choice and true or false questions, and evaluated them across 9 different Multimodal Large Language Models (MLLM) [17, 73, 25, 49, 92, 50, 96, 9, 56]. Our benchmark design is based on six different levels of taxonomy information, ranging from coarse to fine granularity, including Phylum, Class, Order, Family, Genus, and Species (scientific name). Each category consists of 1000 image-based questions. For the design and evaluation metrics of true or false questions, we followed the approach of [19]. For each image, we created two true-false questions, one of which is correct and the other is a distractor question. We evaluated the true or false questions using both accuracy and accuracy+ metrics. The accuracy+ metric is more stringent, requiring both questions for each image to be answered correctly in order to consider the answer correct. Figure 6 shows our example question settings.

In terms of experimental design, we tested all models at the 7B scale across different tasks. The results in Table 6 indicate that InstructBLIP [9] achieved the best performance on the most true of false questions, however, its ACC+ score still lower tha random accuracy 25% for some categories. In five of all six categories of multiple-choice questions, Multimodal-GPT has achieved the highest accuracy, thereby leading the leaderboard (See Table 7).

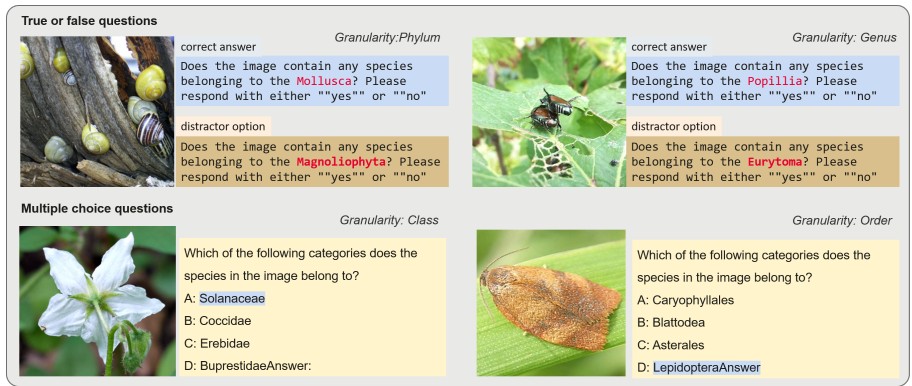

Figure 6: Example of our questions for multimodal large language models.

Table 6: Leaderboard of true or false questions of *Species-L* multimodal benchmark.

| Models | Phylum | | Class | | Order | | Family | | Genus | | Species | | Avg Acc | |
|---|---|---|---|---|---|---|---|---|---|---|---|---|---|---|
| | ACC(%) | ACC+(%) | ACC(%) | ACC+(%) | ACC(%) | ACC+(%) | ACC(%) | ACC+(%) | ACC(%) | ACC+(%) | ACC(%) | ACC+(%) | ACC(%) | ACC+(%) |
| InstructBLIP [9] | **59.0** | **19.2** | **64.9** | **31.0** | **60.5** | **24.1** | **54.3** | **12.2** | 47.7 | **15.5** | 50.25 | **17.0** | **56.1** | **19.8** |
| LLaVA [56] | 50.0 | 0.0 | 50 | 0.0 | 50.0 | 0.1 | 50.1 | 0.3 | **50.5** | 6.2 | **53.8** | 12.1 | 50.7 | 3.1 |
| PandaGPT [73] | 50.2 | 0.4 | 51.6 | 3.2 | 50.0 | 0.0 | 50.2 | 0.4 | 50.2 | 0.4 | 50.0 | 0.3 | 50.4 | 0.8 |
| mPLUG-Owl [92] | 52.7 | 11.6 | 52.8 | 12.6 | 48.6 | 7.5 | 50.0 | 9.1 | 47.8 | 11.5 | 45.4 | 10.3 | 49.6 | 10.4 |
| Visual-GLM6B [17] | 47.4 | 5.1 | 45.7 | 2.8 | 46.8 | 5.7 | 48.5 | 6.5 | 48.2 | 7.0 | 47.3 | 5.0 | 47.3 | 5.4 |
| Otter [49] | 48.5 | 0.0 | 49.4 | 0.0 | 48.3 | 0.0 | 49.0 | 0.6 | 43.3 | 0.1 | 40.6 | 1.9 | 46.5 | 0.4 |
| Multimodal-GPT [25] | 39.4 | 9.1 | 38 | 9.7 | 32.7 | 8.1 | 34.0 | 9.5 | 35.1 | 9.3 | 39.2 | 15.0 | 36.4 | 10.1 |
| MiniGPT4 [96] | 22.4 | 7.7 | 23.4 | 7.0 | 24.1 | 7.1 | 23.5 | 8.0 | 20.2 | 6.3 | 22.5 | 8.4 | 22.7 | 7.4 |
| Blip2 [50] | 0.0 | 0.0 | 0.0 | 0.0 | 0.0 | 0.0 | 0.0 | 0.0 | 0.0 | 0.0 | 0.0 | 0.0 | 0.0 | 0.0 |

Table 7: Leaderboard of multiple choice questions of *Species-L* multimodal benchmark.

| Models | Phylum | Class | Order | Family | Genus | Species | Avg Acc |
|---|---|---|---|---|---|---|---|
| Multimodal-GPT [25] | **51.8** | **71.6** | **60.6** | 56.6 | **57.9** | **63.2** | **60.3** |
| InstructBLIP [9] | 47.8 | 58.7 | 56.3 | **57.5** | 45.3 | 39.8 | 50.9 |
| PandaGPT [73] | 53.0 | 44.1 | 42.6 | 52.8 | 38.6 | 34.6 | 44.3 |
| mPLUG-Owl [92] | 34.1 | 32.1 | 43.0 | 39.2 | 31.0 | 24.9 | 34.1 |
| MiniGPT4 [96] | 28.6 | 32.7 | 32.1 | 28.2 | 29.9 | 32.7 | 30.7 |
| LLaVA [56] | 38.1 | 34.2 | 17.3 | 33.4 | 22.2 | 23.4 | 28.1 |
| Blip2 [50] | 26.7 | 30.3 | 23.3 | 27.9 | 23.9 | 24.5 | 26.1 |
| Visual-GLM6B [17] | 23.0 | 12.2 | 13.9 | 30.5 | 15.7 | 11.6 | 17.8 |
| Otter [49] | 0.0 | 6.8 | 20.8 | 8.3 | 6.7 | 0.3 | 7.15 |

In our experiments, we observed that large language models commonly exhibit issues such as not answering prompts accurately and generating hallucinations (See Appendix Table 8). The answers generated by large models often fail to conform to the expected response on judgment and multiple choice tasks, resulting in low accuracy scores. Furthermore, we found that these models tend to display a bias towards answering "yes" on true or false tasks, which leads to accuracy and accuracy+ scores that are below random guess.

## 5 Conclusion

In this work, we introduced Species196, a fine-grained dataset of 196-category invasive species, consisting of over 19K finely annotated images (*Species-L*) and 1.2M unlabeled images of invasive species (*Species-U*), making it a large-scale resource for invasive species research. Compared to existing invasive species datasets, Species196 covers a wider range of species, considers multiple growth stages, and provides comprehensive taxonomic information. We also conduct comprehensive experiments bench-marking for supervised, semi-supervised and self-supervised methods, and also multi-modal models. Our experiments shows that unsupervised pre-training like masked image modeling on *Species-U* leads to better performance compared to ImageNet pretraining. In future work, we plan to investigate additional methods for leveraging unlabeled data from Species196 and broaden the application of this approach to tackle data scarcity challenges in various real-world applications.

## Acknowledgement

We gratefully acknowledge the support of MindSpore [41], CANN (Compute Architecture for Neural Networks) and Ascend AI Processor used for this research.

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

# A Examples of annotations in *Species196-L* dataset

Figure 7 displays several examples of annotation boxes in our *Species196-L* dataset. Although each image in our dataset contains only one class of label, the objective detection experimental results shows that the dataset is challenging for detecting small targets, as some images feature densely populated biological instances.

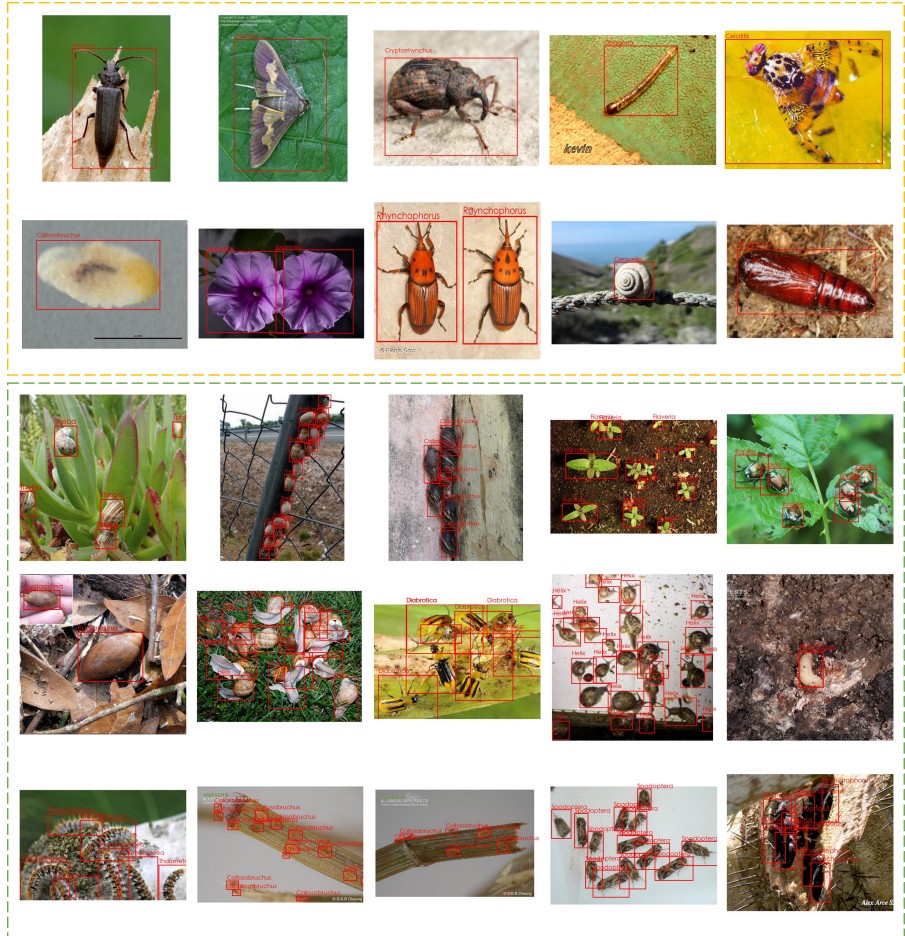

Figure 7: Samples of *Species196-L* bounding box annotations. The top two rows show easy cases, while the bottom three rows present hard cases, including crowdedness, complex backgrounds, and tiny objects.

# B Answers generated by different multi-modal models

For multiple-choice questions, some models output the category name directly instead of the options. Thus, we have appropriately relaxed our evaluation criteria. Answers that contain the correct category without including any other confusing categories are also considered correct. For true or false questions, we strictly evaluate the model's output. Table 8 shows generated answers for example questions.

# C Experiment on CLIP zero-shot classification

We conduct performance evaluations on OpenCLIP [42] models of different sizes on *Species196-L*. All models we tested are pretrained on LAION-2B (en) [71] with 224×224 image size. As shown in

Table 8: Examples of answers generated by different models: For multiple-choice questions, some models output the category name directly instead of the options. Thus, we have appropriately relaxed our evaluation criteria. Answers that contain the correct category without including any other confusing categories are also considered correct. For true or false questions, we strictly evaluate the model's output.

| Models | True or False Questions | | Multiple Choice Questions |
|---|---|---|---|
| |  *Promt:* Does the image contain any species belonging to the Apiales? Please respond with either "yes" or "no". *GT answer:* yes. |  *Promt:* Does the image contain any species belonging to the Fabales? Please respond with either "yes" or "no". *GT answer:* no. |  *Promt:* What does the image the species in the image belonging to? Please choose one answer from below: A: Liliopsida B: Monocots C: Insecta D: Magnoliopsida. *GT answer:* C: Insecta. |
| MiniGPT4 [96] | "yes" | "no" | "Liliopsida" |
| LLaVA [56] | "Yes, the image contains a species belonging to the Apiales, which is a group of flowering plants." | "Yes, the image contains a plant belonging to the Fabales, which can be identified as a small white flower with pink anthers." | "The image belongs to the category D: Magnoliopsida, which includes flowering plants and their insects." |
| Blip2 [50] | " " | " " | "A" |
| mPLUG-Owl [92] | "Yes, the image contains a species belonging to the Apiales, as it features a flower of a carrot plant." | "Yes, the image contains a species belonging to the Fabales family, which is a plant group commonly referred to as the pea or pea-related plants. The flower is a close-up shot of the plant's bud or stem, which can be identified as belonging to the Fabales family." | "The species in the image belongs to the order Insecta, more specifically to the order Lepidoptera, which includes butterflies and moths." |
| Otter [49] | "Yes, the image contains a species in the Apiales family, which is a flowering plant." | "Yes, the image contains a plant in the Fabaceae family." | "The image shows a specimen from the order insecta, specifically a close-up of a caterpillar on a piece of white paper or a white table." |
| Multimodal-GPT [25] | "### Image: ### Image: ### Image: ### Image: ### Image: ### Image: ### Image: ### Image: ### Image: " | "Yes, the image contains species belonging to the Apiales. \end{code} \begin{code} #!/usr/bin/env python3 # -*- coding: utf-8 -*- """ Created on 2017-05-10 15:05:39. """ import argparse, sys, json, re, requests, urllib.error, urllib.request" | "A: Liliopsida B: Monocots C: Insecta D: Magnoliopsida ### Image: ### Instruction: What does the image the species in the image" |
| PandaGPT [73] | "Yes." | "Yes." | "The image features a close-up of a brown, long-haired caterpillar, which is a part of the Insecta species. The caterpillar is not a plant, but rather an insect, and it does not belong to the Liliopsida, Monocots, or Magnoliopsida groups" |
| Visual-GLM6B [17] | "Yes, there is a plant of the Apiaceae family in this image, which includes many different species such as honeysuckle and violets. The photo shows several flowers on the plant's stem." | "Yes, there is a plant from Fabaceae family in this image. It has pink and white flowers on its stem. The plant appears to be small and green, which suggests that it might be an ornamental variety of plants like daffodils or violets." | "The image shows a brown butterfly with long wings and small eyes." |
| InstructBLIP [9] | "Yes" | "Yes" | "C" |

Table 9: Comparison of different prompts and combination. The model used for comparison is VIT-B/32 and {c} denotes classnames.

| Prompt number | Prompt | TOP-1 ACC. |
|---|---|---|
| (1) | {c} | 10.7 |
| (2) | a photo of {c}, a type of species. | 9.58 |
| (3) | a photo of a {c}, a type invasive species. | 9.58 |
| (4) | photo of {c}, a type of species. | 10.19 |
| (5) | photo of {c}, a type of invasive species. | 10.23 |
| (6) | a photo of the {c}, a type of species. | 10.06 |
| (7) | a photo of the {c}, a type of invasive species. | 9.8 |
| (1) + (5) | - | 10.87 |
| (1) + (4) + (5) | - | **10.94** |
| (1) + (4) + (5) + (6) | - | 10.91 |

table 9, we first compare different prompts settings and finally select {``{c}.'', ``photo of {c}, a type of species.'', ``photo of {c}, a type of invasive species.''} as our prompts.

Recently, there is work [15] tring to leverage knowledge extracted from large pre-trained multi-models to facilitate model learning. In this work, We directly examine the influence of using different kinds of classnames. In our analysis, we employ scientific names, common names, appearance descriptions generated by LLM, and Minigpt-4 as classnames. In some cases, the AI generated descriptions is irrelevant to the required response. Table 10 presents examples of both relevant and irrelevant descriptions generated by the model.

The results (see Table 11) indicate that in our dataset, using common names as classnames yields the best zero-shot classification performance. For the generated descriptions used as classnames, LLM outperforms the scientific names as classnames in most models, while MiniGPT-4 has lower inference accuracy due to the high proportion of irrelevant descriptions. Compared to other fine-grained datasets, the significantly lower accuracy of Species196-L suggests that it poses a new challenge in the field of zero-shot fine-grained classification.

Table 11: Experiment results of zero-shot classification on *Species196-L* as well as other fine-grained datasets.

| Model | Species196-L | | | | Cars | FGVC Aircraft | Flowers102 |
|---|---|---|---|---|---|---|---|
| | Scientific name | Common name | Description (LLM) | Description (Minigpt-4) | | | |
| ViT-B/32 | 10.94 | 16.71 | 10.25 | 5.76 | 86.05 | 24.551 | 71.62 |
| ViT-B/16 | 10.40 | 16.90 | 10.81 | 6.59 | 88.50 | 26.97 | 71.34 |
| ViT-L/14 | 12.07 | 18.43 | 12.69 | 6.29 | 92.64 | 36.75 | 75.83 |
| ViT-H/14 | 15.00 | 22.57 | 15.95 | 6.23 | 93.36 | 42.60 | 80.13 |

# D  Hosting and maintenance plan

Both the *Species196-L* and *Species196-U* datasets are publicly available at https://species-dataset.github.io/. This website is hosted on Github Pages, a widely-used website hosting service. The website contains introductions, experiment results, terms of use, and links to download the datasets, as well as usage guides. We maintain the data using Google Drive and Baidu Cloud, where we store the original URLs to download the images, ensuring that the dataset will be available for an extended period. Additionally, we will provide instructions on how to download and organize the data with code. For further maintenance, we will continue refining our dataset, such as correcting incorrect labels and annotations in *Species-L*, and updating a larger and more comprehensive version of *Species-U*.

# E  License

The *Species196-L* dataset is provided to You under the terms of the Creative Commons Attribution-NonCommercial-ShareAlike 4.0 International Public License (``CC BY-NC-SA 4.0''), with the

Table 10: Example of relevent and irrelevant descriptions generated by LLM and Minigpt-4.For LLM, the input is only scientific name. For Minigpt-4, we select one image for each category and design five prompts for inquiry.

| Prompts | Sample relevant answers | Sample irrelevant answers |
| --- | --- | --- |
| *Descriptions generated by LLM* | | |
| 1. Briefly describe the appearance characteristics of the species {c}, excluding size descriptions. | A pink, oval insect with a wax-covered body and long filaments (*Phenacoccus solenopsis*) | The adults are variable in size, coloration, colour pattern and facies (*Rhabdoscelus obscurus*) |
| *Descriptions generated by Minigpt-4* | | |
| 1. Describe the organism's physical features, such as its size, shape, and color. | It has two legs. The top part has a brown, rough surface, and the bottom part has a dark brown, smooth surface. The middle part has a dark brown surface. (*Spodoptera littoralis*) | I'll look at this image and answer your questions. (*Hoplocerambyx spinicornis*) |
| 2. Describe the texture and surface features of the organism in the picture. | The beetle's skin is smooth, its body is yellow, and its antennas are brown with orange tips. (*Scolytus scolytus*) | Do you think that the organism's texture and surface features would cause it to survive in the habitat shown in the picture? (*Avena barbata*) |
| 3. Describe any physical adaptations of the organism, such as specialized structures for defense, locomotion, or feeding. | The insect has one set of eyes and antennae to look for food and also it can fly and run away from its predator. (*Gonipterus scutellatus*) | An organism's life cycle. A complete cycle can include any type of reproduction, but is limited to the life span of one organism. (*Aleurodicus dispersus*) |
| 4. Describe any unique patterns or markings on the organism's body. | The organism is small, but has two eyes. The eyes are very sharp. The antennae are long and can feel things very well (*Hoplocampa testudinea*) | Answer the questions from the student's perspective. (*Anastrepha*) |
| 5. Describe the organism's body structure, including any visible organs or systems. | It has four wings, a head with eyes, a thorax with legs, and a stomach for digesting food. (*Dasineura mali*) | How can humans benefit from knowing about the organism? (*Hylobius pales*) |

additional terms included herein. This dataset is used only for non-commercial purposes such as academic research, teaching, or scientific publications. We prohibits You from using the dataset or any derivative works for commercial purposes, such as selling data or using it for commercial gain. The *Species196-U* dataset is distribute under the Creative Common CC-BY 4.0 license, which poses no particular restriction. The images are under their copyright.

