# OpenReview forum: "Species196: A One-Million Semi-supervised Dataset for Fine-grained Species Recognition"
_NeurIPS.cc/2023/Track/Datasets_and_Benchmarks — NeurIPS 2023 Datasets and Benchmarks Poster_

### Official Review · Reviewer_fB1t · 2023-07-05
**This paper proposes Species196, a large-scale semi-supervised dataset of 196-category invasive species.**

**Rating:** 6
**Confidence:** 4
**Clarity:** Yes.

**Strengths:**

1.	The paper is generally easy to read and easy to follow.
2.	The paper states that bounding boxes were labeled for insect pests in accordance with the COCO format. This indicates that the dataset includes spatial information and enables sample-aware recognition, which can be beneficial for accurate pest location and categorization tasks.
3.	The establishment of a hierarchical taxonomy system, spanning from domain to species level, provides detailed classification information for each invasive species. This can be valuable for researchers and practitioners working in the field of invasive species identification and management.
4.	The Species196-L dataset includes images of invasive species at various life stages, such as eggs, larvae, pupae, and adults. This multi-grained approach allows for the study of different developmental stages and their impact on ecosystems and economies.
5.	The article provides a wealth of baseline experiments, offering significant assistance to other researchers in furthering this research endeavor.


**Additional Feedback:**

None.

**Correctness:**

The dataset is constructed in a sound way. Authors also provide extensive experiments.

**Documentation:**

Authors provide enough details on data collection and organization.

**Ethics:**

This article does not raise any ethical concerns.

**Limitations:**

I think authors have mentioned the limitations and potential negative societal impact of their work.

**Opportunities For Improvement:**

1.	While the author has provided numerous baseline experimental results, they are limited to basic models. From a methodological perspective, the article lacks extensive experiments on fine-grained techniques, which are necessary for the continuation of this work. Based on the current provided experimental results, researchers could easily employ more powerful models to address the problem discussed in the paper, which is not expected.
2.	The paper lacks comparative experiments with semi-supervised and other related method. One of the main contributions of the article is the introduction of the 'Species196-U' dataset, which consists of 1.2 million unlabeled data. However, the article does not provide direct evidence to demonstrate that these data can generate positive feedback based on the 'Species196-L' dataset in a fair experimental environment.


**Relation To Prior Work:**

Yes.

**Summary And Contributions:**

This article presents the creation of two datasets, Species196-L and Species196-U, for invasive species recognition. Species196-L focuses on insects, weeds, and mollusks and contains 19,236 images of 196 invasive species. The dataset includes a comprehensive taxonomy system and was collected from various online sources. Species196-U is a large-scale dataset obtained from LAION-5B using the clip-retrieval method, containing around 1.5 million unlabeled images of 196 insect and weed species. The challenges of the datasets include class similarity, low variability, and data imbalance. The article emphasizes the importance of accurate identification and location of invasive organisms for real-world applications.

---

> ### Author Response · Authors · 2023-08-22
> **Review response for fB1t, Part 1**
>
> ### Opportunities For Improvement
>
> > O-1: While the author has provided numerous baseline experimental results, they are limited to basic models. From a methodological perspective, the article lacks extensive experiments on fine-grained techniques, which are necessary for the continuation of this work. Based on the current provided experimental results, researchers could easily employ more powerful models to address the problem discussed in the paper, which is not expected.
>
> We appreciate your feedback regarding the scope of our experiments. In this work, we have provided baseline results using basic supervised, semi-supervised, and self-supervised models to offer a preliminary assessment of the dataset's characteristics and potential challenges.
>
> For fine-grained techniques, we conducted experiments using Metaformer, TransFG, and IELT. Notably, the Metaformer model showed excellent performance on the *Species196-L* dataset.
>
> While the primary goal of this study was to introduce the new dataset, we acknowledge that the exploration of fine-grained techniques would be a valuable addition to this work. In response to this, we have conducted additional experiments using models which have previously shown good performance on the [CUB-200-2011 dataset leaderboard](https://paperswithcode.com/sota/fine-grained-image-classification-on-cub-200) and with public available official code.
>
> Please note that our experimental hyperparameters, resolution, and pre-training strategies are based on the settings from the official code repositories of these models. Therefore, the comparisons may not be absolutely strict.
>
> The results of these additional experiments are as follows:
>
> | Model                                             | Backbone     | Resolution | Pretrain       | Top1 Acc |
> | :------------------------------------------------ | ------------ | ---------- | -------------- | :------- |
> | *Metaformer*-2                                    | Metaformer-2 | *384x384*  | *Imagenet-22K* | 88.69    |
> | *Metaformer*-2                                    | Metaformer-2 | *224x224*  | *Imagenet-22K* | 87.69    |
> | [SIM-Trans ↗](https://arxiv.org/abs/2208.14607v1) | ViT-B_16     | 448x448    | *Imagenet-22K* | 85.20    |
> | [ViT-NeT ↗](https://github.com/jumpsnack/ViT-NeT) | Swin-B       | 448x448    | *Imagenet-22K* | 84.73    |
> | *TransFG*                                         | ViT-L_16     | *224x224*  | *Imagenet-22K* | 84.42    |
> | *IELT*                                            | ViT-B_16     | *448x448*  | *Imagenet-22K* | 81.92    |
> | [HERBS ↗](https://arxiv.org/abs/2303.06442v2)     | Swin-T       | 384x384    | *Imagenet-22K* | 81.48    |
> | [PIM ↗](https://arxiv.org/abs/2202.03822v1)       | Swin-T       | 384x384    | CUB-200-201    | 77.80    |
>
>
>
> We hope these additional experiments help to address your concerns to some extent and provide a more comprehensive benchmark for fine-grained techniques. We will also continue to explore the application of novel fine-grained classification methods on *Species196*.

---

> ### Author Response · Authors · 2023-08-22
> **Review response for fB1t, Part 2**
>
> ### Opportunities For Improvement
>
> > O-2: The paper lacks comparative experiments with semi-supervised and other related method. One of the main contributions of the article is the introduction of the 'Species196-U' dataset, which consists of 1.2 million unlabeled data. However, the article does not provide direct evidence to demonstrate that these data can generate positive feedback based on the 'Species196-L' dataset in a fair experimental environment.
>
> Thank you for your insightful suggestions. In response, we have expanded our experimental scope to include popular semi-supervised methods such as Mixmatch, FixMatch, and RemixMatch. In these experiments, we treated *Species196-L* as labeled data and *Species196-U* as unlabeled data. The results of these experiments using ResNet50 are shown below, indicating that our *Species196-U* dataset plays a role in these semi-supervised learning approaches:
>
> | Models   | Method     | Top1 ACC(only Species-L) | Top1 ACC(semi-supervised using Specis-U) |
> | :------- | :--------- | :----------------------: | :--------------------------------------: |
> | Resnet50 | Mixmatch   |          64.32           |                  67.87                   |
> | Resnet50 | Fixmatch   |          64.32           |                  66.94                   |
> | Resnet50 | Remixmatch |          64.32           |                  69.30                   |
>
> Additionally, we conducted experiments using contrastive learning methods such as Swav and MoCo v2. The linear probe results of *Species196-L* are as follows:
>
> | Models   | Method  | Batch Size | Pretrain Data | Pretrain Epoch | Linear_cls Epoch | Top-1 ACC |
> | :------- | :------ | :--------- | :------------ | :------------- | :--------------- | :-------- |
> | Resnet50 | swav    | 256        | In1k          | 200            | 100              | 69.0      |
> | Resnet50 | swav    | 256        | Species-U     | 200            | 100              | 69.7      |
> | Resnet50 | MoCo v2 | 256        | In1k          | 800            | 100              | 67.1      |
> | Resnet50 | MoCo v2 | 256        | Species-U     | 800            | 100              | 67.9      |
>
> The results demonstrate that incorporating unlabeled data from *Species196-U* leads to a noticeable improvement in performance in both semi-supervised and contrastive learning settings.
>
> Regarding your second point about the lack of direct evidence in our article to demonstrate the positive feedback generated by the *Species196-U* dataset based on the *Species196-L* dataset, we believe that we have addressed this in the *4.2.2 Masked image modeling* section of our paper, specifically in Table 5. The results of the Swin-B, Convnext-T, and Convnext-B models, when pretrained with *Species-U* using unsupervised masked image modeling, show a significant improvement compared to the results obtained with unsupervised pretraining using Imagenet-1K. Moreover, these improvements are achieved using the same scale of pretraining data, employing the same pretraining methods, and with the same number of epochs for both the pretraining and finetuning processes. So we think it's a fair experiment to show that *Species196-U* can generate positive feedback based on the *Species196-L* dataset.

---

> > ### Comment · Reviewer_fB1t · 2023-08-30
> >
> > Thank you for the response. I would like to recommend accepting this paper.

---

> > > ### Author Response · Authors · 2023-08-30
> > >
> > > Dear Reviewer,
> > >
> > > Thank you for your recommendation to accept our paper. We appreciate your positive feedback. Be assured, we will continue to enhance and update our *Species196* dataset.
> > >
> > > Best regards,
> > > Author

---

### Official Review · Reviewer_H3At · 2023-07-20
**Review of Species196**

**Rating:** 8
**Confidence:** 3
**Clarity:** The paper is written fairly well.

**Strengths:**

The paper presents a well crafted dataset, consisting of images with expert annotations of invasive species, accompanied with taxonomical information regarding the life stage of the pictured species. The combination with the large scale unlabeled Species196-U dataset, makes this a strong contribution to the fine-grained image analysis community, and very relevant. The datasets are thoroughly evaluated across several tasks.

**Additional Feedback:**

There are a few places with repetition e.g.:
1) Line 76-79, description of the deep learning's dependency on high quality annotated data.
1) Line 207-212, describing the Metaformer results.

In Table 2 MaxViT-T is noted with a * instead of a dagger.

**Correctness:**

The dataset construction and experiments are in general sound but there are some points to be addressed:

1) In line 146-147 it is stated that the dataset is split 1:1 into a training and test split. Why did you choose to not establish a validation split, in order to not overfit to the test split? This seems like a major oversight.
2) It is unclear how many of the images in the Species196-L dataset is bounding box annotated. Just a subset (L157-158) or all (L161-162)?
3) Why only use three images from Species196-L when sampling 8000 images per class for Species196-U? This seems like it would cause a large bias.
4) It is unclear what is meant by the 120000 text-image pair remark in Line 180. Could you elaborate?
5) Did you visually verify the Species196-U, to ensure that the sampled images would match the intended class?
6) In Line 182-183 it is stated that the dataset is like other fine grained dataset. Why not measure this using the method proposed in the paper "Measuring Dataset Granularity" by Cui et al.?
7) It is unclear which type of accuracy is use in the benchmarks. Are you using the micro accuracy (i.e. average across all samples, favoring dominant classes) or macro accuracy (i.e. average per class accuracy, assigning equal weight to all classes)
8) It is unclear at which IoU threshold the object detection benchmark metrics are measured at.
9) It is unclear what kind, if any, hyperparameter searches was performed for the benchmarks.

**Documentation:**

The paper documents the data collection process, availability and maintenance plans, and licensing.

**Ethics:**

No.

**Limitations:**

There are not any immediate limitations of the presented work.

**Opportunities For Improvement:**

The dataset may be a bit small for modern purposes, as it only consists of 19K annotated images, compared to larger scale fine grained datasets such as Herbarium-19, PlantNet, and iNaturalist. There are also a set of considerations regarding the dataset assembly and benchmarks, see below.

**Relation To Prior Work:**

The paper positions itself well against the related works, and discuss many relevant previous works.

**Summary And Contributions:**

The paper proposes two new datasets, Species196-L and Species196-U, for classification of 196 invasive species. The  two datasets differ in how the data is collected and annotated. Species196-L contain 19K expert annotated images, whereas Species196-U contains 1.2 million images from the LAION-5B dataset, selected based on similarity to the classes of Species196-L but without labels.
The paper contains several benchmarks, comparing classification performance when training just on Species196-L, as well as pretraining with Species196-U. Object detection, as well as semi and self supervised benchmarks are also presented.

---

> ### Author Response · Authors · 2023-08-22
> **Review response for H3At, Part 1**
>
> ### Opportunities For Improvement:
>
> > O-1:  The dataset may be a bit small for modern purposes, as it only consists of 19K annotated images, compared to larger scale fine grained datasets such as Herbarium-19, PlantNet, and iNaturalist.
>
> Thank you for your insightful comment. We agree that large-scale datasets such as Herbarium-19, PlantNet, and iNaturalist have their value in the field of fine-grained classification. However, we would like to highlight a few unique aspects of our Species196 dataset:
>
> - **Focused Application**: Species196 is specifically designed to address real-world invasion biometrics problems. This makes it particularly important and unique in its field of application.
>
> - **Quality Assurance**: Despite its smaller size compared to the datasets you mentioned, our images have undergone manual discernment and annotation to ensure their quality and correctness. We have prioritized quality over quantity in the construction of our dataset.
>
> - **Challenging Testbed for Real-world Applications**: We believe that the value of our dataset lies in its ability to serve as a challenging testbed for real-world applications. It offers an opportunity to explore how to leverage large-scale unlabeled data and improve model performance on imbalanced, fine-grained data.
>
> We appreciate your feedback and will continue our efforts to expand and enhance our dataset. We hope that the unique characteristics and potential applications of our Species196 dataset are taken into consideration.
>
> ### Correctness:
>
> > C-1: In line 146-147 it is stated that the dataset is split 1:1 into a training and test split. Why did you choose to not establish a validation split, in order to not overfit to the test split? This seems like a major oversight.
>
> Thank you for pointing out your concern regarding our dataset split. As stated in line 145, page 5 of our paper, our decision to split the dataset 1:1 into a training and test set follows the approach of Stanford Cars and CUB200. We didn't separately establish a validation set for the following reasons:
>
> We believe that researchers using our Species196 dataset have the flexibility to further split the training set into a validation set as per their specific requirements. This allows them to set up their validation process in a way that best fits their experimental design and avoids overfitting.
>
> >C-2: It is unclear how many of the images in the Species196-L dataset is bounding box annotated. Just a subset (L157-158) or all (L161-162)?
>
> Thank you for your query. We apologize for any confusion caused by the discrepancy between line 157-158 and 161-162 in our paper. To clarify, every image in the Species196-L dataset does indeed have bounding box annotations, contrary to what is suggested in lines 157-158.
>
> We acknowledge the error in our manuscript and appreciate your attention to detail. We will correct this in the revised version and make it clear that all 19236 images in the Species196-L dataset come with bounding box annotations.
>
> We hope this clarification addresses your concern, and we thank you again for your valuable feedback.
>
> > C-3: Why only use three images from Species196-L when sampling 8000 images per class for Species196-U? This seems like it would cause a large bias.
>
> Thank you for your question. The decision to use only three images from each class in Species196-L when sampling for Species196-U was driven by a balance of considerations.
>
> During the retrieval process, we observed significant overlap among the images retrieved for the same class. This implies that using all images from each class for retrieval may not be necessary to capture the representative features of that class. Additionally, using the entire set of 19,236 images for retrieval would increase the computational cost of retrieval, deduplication, and downloading by several orders of magnitude.
>
> While we understand the concern about potential bias, we believe this strategy offered a reasonable trade-off. It allowed us to leverage the labeled data and foster diversity in the unlabeled data while managing computational resources costs. We appreciate your feedback and will take it into account in future work.
>
> > C-4: It is unclear what is meant by the 120000 text-image pair remark in Line 180. Could you elaborate?
>
> Thank you for pointing out this issue. We apologize for the typographical error in line 180 of our manuscript. The correct number of text-image pairs in our dataset is indeed 1,200,000, not 120,000 as originally stated. We appreciate your attention to detail and will correct this error in the revised version of our paper. Thank you again for your valuable feedback.

---

> ### Author Response · Authors · 2023-08-22
> **Review response for H3At, Part 2**
>
> ### Correctness:
>
> > C-5: Did you visually verify the Species196-U, to ensure that the sampled images would match the intended class?
>
> Thank you for your question. Even though the large volume of images in the Species196-U dataset makes manual verification of each image impractical, we have taken steps to visually inspect a subset of the data. To validate this, we conducted a visual inspection of images retrieved at various similarity ranks (100, 500, 1000, and 5000). This is illustrated in Appendix Figure 2 of our paper, where we present examples of these images. Even at a rank as high as 5000, the retrieved images still exhibit a substantial resemblance to the original species image, indicating a reasonable level of class consistency within the dataset.
>
> Our approach to constructing the Species196-U dataset was designed with the understanding that not every sampled image would perfectly match the intended class. However, we aimed to ensure a level of visual similarity that would make the dataset valuable for semi-supervised and unsupervised learning scenarios.+
>
> In summary, while we did not manually verify each image due to the dataset size, we did perform visual inspections to confirm that the Species196-U dataset contains images that are visually similar. We hope this addresses your question, and we appreciate your careful review of our work.
>
> > C-6: In Line 182-183 it is stated that the dataset is like other fine grained dataset. Why not measure this using the method proposed in the paper "Measuring Dataset Granularity" by Cui et al.?
>
> Thank you for your insightful comments and the suggestion to quantify the granularity of our dataset using the method proposed by Cui et al. in "Measuring Dataset Granularity".
>
> We acknowledge the potential value of this method. However, at the time of our analysis, the implementation of this method was not publicly available, which made it impossible for us to directly apply their methodology to our dataset.
>
> In response to your suggestion, we have utilized some related work mentioned in "Measuring Dataset Granularity" to evaluate our dataset. Specifically, we calculated several supervised and unsupervised metrics, including NMI, Rand Index, FMI, Silhouette score, Calinski-Harabasz score, and Davies-Bouldin score. These metrics were used to compare our *Species196-L* dataset with Cifar100 and Inat2021-mini-val). Features were extracted using a pretrained ResNet50 model from the PyTorch library, and clustering was performed using k-means.
>
> The results are as follows:
>
> | Metrics                 | Metrics Type | Cifar100  | Inat2021-mini | SPecies196-L |
> | ----------------------- | ------------ | --------- | ------------- | ------------ |
> | NMI                     | Supervised   | 0.06792   | 0.00185       | *0.00178* |
> | Rand Index              | Supervised   | 0.00559   | -0.25609      | -*1.00513* |
> | FMI                     | Supervised   | 0.01564   | 0.01209       | *0.01426*  |
> | Silhouette score        | Unsupervised | -0.00422  | -0.01135      | *0.01194* |
> | Calinski-Harabasz score | Unsupervised | 110.94859 | 127.32491     | *34.06678* |
> | Davies-Bouldin score    | Unsupervised | 4.62351   | 4.54351        | *3.84286*  |
>
> From these metrics, we observe that our Species196-L dataset shows lower scores than the comparative datasets in terms of clustering, indicating a finer granularity. This supports our assertion about the fine-grained nature of our dataset.

---

> ### Author Response · Authors · 2023-08-22
> **Review response for H3At, Part 3**
>
> ### Correctness:
>
> > C-7: It is unclear which type of accuracy is use in the benchmarks. Are you using the micro accuracy (i.e. average across all samples, favoring dominant classes) or macro accuracy (i.e. average per class accuracy, assigning equal weight to all classes)
>
> Thank you for pointing out the ambiguity in our paper. To clarify, the accuracy reported in our benchmark results is the micro accuracy, which is computed as the average across all samples and favors dominant classes. We chose to use micro accuracy because it is a common metric in image classification tasks and allows for straightforward comparison with other studies.
>
> However, we understand the importance of considering class imbalance, especially in a dataset like ours. Therefore, we now provide the Macro F1 score in our benchmark, which assigns equal weight to all classes regardless of their size.
>
> Here are the updated benchmark results. In the newly added "Macro-F1 score" column, the values before the slash (`/`) represent the Macro F1 score from scratch, and the values after the slash (`/`) represent the Macro F1 score with ImageNet-1K pretraining:
>
> | Model                 | Resolution | Params | FLOPs  | Top-1 ACC     | Top-5 ACC     | Macro-F1 score |
> | :-------------------- | :--------- | :----- | :----- | :------------ | :------------ | -------------- |
> | MobileViT-XS          | 224x224    | 2.3 M  | 0.7 G  | 64.11 / 78.55 | 83.51 / 91.92 | 53.52 / 69.01  |
> | GhostNet 1.0          | 224x224    | 5.2 M  | 0.1 G  | 62.75 / 76.02 | 82.58 / 90.77 | 51.30 / 64.93  |
> | EfficientNet-B0       | 224x224    | 5.3 M  | 0.4 G  | 62.88 / 78.26 | 81.66 / 91.60 | 53.13 / 66.91  |
> | MobileNetV3 Large 1.0 | 224x224    | 5.4 M  | 0.2 G  | 62.75 / 77.83 | 81.46 / 90.77 | 49.99 / 66.50  |
> | RegNetY-4GF           | 224x224    | 20.6 M | 4.0 G  | 43.01 / 82.25 | 69.02 / 93.71 | 28.99 /  71.24 |
> | Deit-S                | 224x224    | 22 M   | 4.6 G  | 36.89 / 77.21 | 56.79 / 91.52 | 29.35 / 65.25  |
> | TNT-S                 | 224x224    | 23.8 M | 5.2 G  | 38.66 / 80.67 | 59.14 / 93.17 | 30.67 / 69.34  |
> | CMT-S                 | 224x224    | 25.1 M | 4.0 G  | 40.86 / 81.12 | 60.10 / 93.32 | 33.25 / 70.40  |
> | Resnet50              | 224x224    | 25.6 M | 4.1 G  | 64.32 / 78.11 | 81.70 / 91.91 | 53.31 / 67.29  |
> | Swin-T                | 224x224    | 28 M   | 4.5 G  | 46.88 / 81.66 | 68.57 / 93.52 | 37.30 / 71.20  |
> | Convnext-T            | 224x224    | 29M    | 4.5 G  | 46.36 / 78.94 | 68.59 / 92.44 | 37.16 /  70.43 |
> | MaxViT-T              | 224x224    | 31 M   | 5.6 G  | 52.19 / 83.35 | 72.12 / 94.16 | 42.40 / 62.56  |
> | MViTv2-B              | 224x224    | 52 M   | 10.2 G | 46.22 / 83.79 | 66.21 / 94.81 | 35.83 / 72.94  |
> | Resnet200-D           | 224x224    | 65 M   | 26.0 G | 51.35 / 82.11 | 73.07 / 94.76 | 37.70 / 70.61  |
> | VIT-B/32              | 224x224    | 86 M   | 8.6 G  | 32.59 / 74.68 | 53.76 / 89.76 | 25.20 / 63.38  |
> | Swin-B                | 224x224    | 88 M   | 15.4 G | 48.72 / 82.88 | 69.71 / 94.30 | 39.28 / 72.04  |
>
> We appreciate your insightful comment and will make sure to clarify the type of metrics we used and update the benchmark with Macro F1 score in the revised version of our paper and our website.
>
> > C-8: It is unclear at which IoU threshold the object detection benchmark metrics are measured at.
>
> We apologize for any confusion caused by the lack of explicit clarification in our paper. For our object detection benchmark metrics, we use the mean Average Precision (mAP) calculated at an Intersection over Union (IoU) threshold range of 0.5 to 0.95 (mAP@0.5:0.95). This is a common practice in evaluating object detection models.
>
> We appreciate your careful attention to detail and thank you for pointing out this omission. We will ensure to provide clear and specific descriptions of our evaluation metrics in our revised version.

---

> ### Author Response · Authors · 2023-08-22
> **Review response for H3At, Part 4**
>
> ### Correctness:
>
> > C-9: It is unclear what kind, if any, hyperparameter searches was performed for the benchmarks.
>
> In our benchmarks, we utilized the `timm` library for model evaluations. For a fair comparison among similar models, such as EfficientNet-B0 and GhostNet 1.0, or Swin-T and TNT-S, we applied similar hyperparameters from what is suggested in timm's official scripts and other common settigs for training particular types of model. We aimed for a reasonable comparison, rather than an extensive hyperparameter search for each model.
>
> "Except for what is mentioned in lines 199-120 on page 7, such as input resolution and training epochs, we also use other important hyperparameters and settings. These are separately listed for convolution-based networks (Convs) and transformer-based networks (Trans)：
>
> | Hyperparameter | Small Convs | Small Trans | Medium Convs | Medium Trans | Large Convs | Large Trans |
> | -------------- | ----------- | ----------- | ------------ | ------------ | ----------- | ----------- |
> | device         | 8 V100s     | 8 V100s     | 8 V100s      | 8 V100s      | 8 V100s     | 8 V100s     |
> | amp            | True        | True        | True         | True         | True        | True        |
> | batch_size     | 256         | 256         | 256          | 256          | 128         | 128         |
> | decay_rate     | 0.1       | 0.1         | 0.1          | 0.1          | 0.1         | 0.1         |
> | lr             | 0.1         | 0.1         | 0.08         | 0.08         | 0.06        | 0.06        |
> | momentum       | 0.9         | 0.9         | 0.9          | 0.9          | 0.9         | 0.9         |
> | opt            | sgd         | sgd         | sgd          | sgd          | sgd         | sgd         |
> | reprob         | 0.2         | 0.4         | 0.2          | 0.4          | 0.2         | 0.4         |
> | lr_sched       | cosine      | cosine      | cosine       | cosine       | cosine      | cosine      |
> | warmup_epochs  | 5           | 15          | 5            | 15           | 5           | 15          |
> | weight_decay   | 0.01        | 0.05        | 0.01         | 0.05         | 0.01        | 0.05        |

---

> > ### Comment · Reviewer_H3At · 2023-08-29
> > **Updated score**
> >
> > Thank you for the thorough response(s) to my own and the other reviewers remarks. All of my comments have been addressed in a generally satisfactory manner, leading me to increase my score to 8. I believe the dataset is very interesting and the authors have gone through a lot more of effort to provide extra results and explanations.
> >
> > My only remaining comment is in regards of C-1 (no Val split.) While I can follow the reasoning of the authors, I would like to encourage the authors to release the exact train and val split they used for the experiments in order to allow for direct comparisons.

---

> > > ### Author Response · Authors · 2023-08-29
> > >
> > > Dear Reviewer,
> > >
> > > Thank you for your constructive feedback and the improved score. In response to your suggestion, we will definitely release *Species196-L*'s train-validation split to facilitate direct comparisons for researchers. We appreciate your valuable advice.
> > >
> > > Best regards,
> > > Author

---

### Official Review · Reviewer_foZN · 2023-07-20
**A benchmark dataset with a potential**

**Rating:** 7
**Confidence:** 4
**Clarity:** The reported findings and the way the…

**Strengths:**

1.	The introduction of a new large dataset for the development of machine learning approaches devoted to species of Mollusca, weeds, and insects recognition.
2.	Results of a large number of representative methods for future reference.

**Additional Feedback:**

Please address the issues related to the maintenance plan and the reported performance criteria.

**Correctness:**

The mentioned discussion on the usage of certain performance criteria may lead to a correction of the results.

**Documentation:**

The availability and maintenance of the benchmark require additional effort as mentioned in Limitations.  The documentation lacks hosting and maintenance plan.

**Ethics:**

There are no ethical concerns related to this study.

**Limitations:**

1.	The study reports results of performance indices that may be affected by classes with a higher number of samples.
2.	As the labeling of bounding boxes for pest locations was carried out on randomly selected subsets of images, their quantity should be reported somewhere.
3.	The download links at https://species-dataset.github.io/download.html are not permanent and appear to be more like personal (Google Drive) storage. Therefore, the availability and maintenance of the benchmark require serious attention.

**Opportunities For Improvement:**

1.	The results shown in the paper are meant to be reported by other researchers that will use the images for the development of their methods. However, due to the  imbalance of the dataset , certain performance metrics should not be employed and might be skipped by other researchers. Please discuss the suitability of used performance criteria.
2.	T The paper's title mentions "recognition," yet the abstract refers to a broader usage of the benchmark, encompassing more than just classification (recognition).
3.	Four experimental settings are mentioned in abstract but only three are explicitly enlisted (supervised learning, semi-supervised learning, and self-supervised pretraining). Please clarify.
4.	Surprisingly, the masked image modeling is first mentioned in Section 4.4.2. A reader would not expect this part to be included in the study. Is it important? Why was it included?
5.	 The paper requires thorough proofreading and correction of grammar issues, e.g., see reference [1] or “faces. we” (Section 3.2)

**Relation To Prior Work:**

The limitations of previous studies are discussed.

**Summary And Contributions:**

In the paper, a benchmark dataset for the development of machine learning approaches dedicated to the recognition of  of Mollusca, weeds, and insects is introduced.
Contributions of the study: 1) a labeled large-scale dataset, 2) numerous experiments with methods from closely related machine learning fields.

---

> ### Author Response · Authors · 2023-08-22
> **Review response for foZN, Part 1**
>
> ### Opportunities For Improvement:
>
> > O-1: The results shown in the paper are meant to be reported by other researchers that will use the images for the development of their methods. However, due to the imbalance of the dataset , certain performance metrics should not be employed and might be skipped by other researchers. Please discuss the suitability of used performance criteria.
>
> Thank you for your comment on our choice of performance metrics. We agree that the Top-1 accuracy may not fully capture the performance of a model on an imbalanced dataset.
>
> In response to this feedback, we will supplement our main benchmark evaluation with macro F1 scores in the revised version of our paper. Macro F1 score (taking all classes as equally important) is a commonly used metric for multi-class classification problems, particularly when the classes are imbalanced.
>
> The F1 Score formula can be represented as:
> $$
> \text{F1 Score} = 2 \times \frac{{(\text{Precision} \times \text{Recall})}}{{\text{Precision} + \text{Recall}}}
> $$
> The Macro F1 Score formula can be represented as:
> $$
> \text{Macro F1 Score} = \frac{{\text{F1 Score Class 1} + \text{F1 Score Class 2} + \ldots + \text{F1 Score Class N}}}{N}
> $$
> Thank you for your suggestion. We believe the Macro F1 score will be particularly beneficial in evaluating our dataset. Here are the updated benchmark results. In the newly added "Macro-F1 score" column, the values before the slash (`/`) represent the Macro F1 score from scratch, and the values after the slash (`/`) represent the Macro F1 score with ImageNet-1K pretraining:
>
> | Model                 | Resolution | Params | FLOPs  | Top-1 ACC     | Top-5 ACC     | Macro-F1 score |
> | :-------------------- | :--------- | :----- | :----- | :------------ | :------------ | -------------- |
> | MobileViT-XS          | 224x224    | 2.3 M  | 0.7 G  | 64.11 / 78.55 | 83.51 / 91.92 | 53.52 / 69.01  |
> | GhostNet 1.0          | 224x224    | 5.2 M  | 0.1 G  | 62.75 / 76.02 | 82.58 / 90.77 | 51.30 / 64.93  |
> | EfficientNet-B0       | 224x224    | 5.3 M  | 0.4 G  | 62.88 / 78.26 | 81.66 / 91.60 | 53.13 / 66.91  |
> | MobileNetV3 Large 1.0 | 224x224    | 5.4 M  | 0.2 G  | 62.75 / 77.83 | 81.46 / 90.77 | 49.99 / 66.50  |
> | RegNetY-4GF           | 224x224    | 20.6 M | 4.0 G  | 43.01 / 82.25 | 69.02 / 93.71 | 28.99 /  71.24 |
> | Deit-S                | 224x224    | 22 M   | 4.6 G  | 36.89 / 77.21 | 56.79 / 91.52 | 29.35 / 65.25  |
> | TNT-S                 | 224x224    | 23.8 M | 5.2 G  | 38.66 / 80.67 | 59.14 / 93.17 | 30.67 / 69.34  |
> | CMT-S                 | 224x224    | 25.1 M | 4.0 G  | 40.86 / 81.12 | 60.10 / 93.32 | 33.25 / 70.40  |
> | Resnet50              | 224x224    | 25.6 M | 4.1 G  | 64.32 / 78.11 | 81.70 / 91.91 | 53.31 / 67.29  |
> | Swin-T                | 224x224    | 28 M   | 4.5 G  | 46.88 / 81.66 | 68.57 / 93.52 | 37.30 / 71.20  |
> | Convnext-T            | 224x224    | 29M    | 4.5 G  | 46.36 / 78.94 | 68.59 / 92.44 | 37.16 /  70.43 |
> | MaxViT-T              | 224x224    | 31 M   | 5.6 G  | 52.19 / 83.35 | 72.12 / 94.16 | 42.40 / 62.56  |
> | MViTv2-B              | 224x224    | 52 M   | 10.2 G | 46.22 / 83.79 | 66.21 / 94.81 | 35.83 / 72.94  |
> | Resnet200-D           | 224x224    | 65 M   | 26.0 G | 51.35 / 82.11 | 73.07 / 94.76 | 37.70 / 70.61  |
> | VIT-B/32              | 224x224    | 86 M   | 8.6 G  | 32.59 / 74.68 | 53.76 / 89.76 | 25.20 / 63.38  |
> | Swin-B                | 224x224    | 88 M   | 15.4 G | 48.72 / 82.88 | 69.71 / 94.30 | 39.28 / 72.04  |
>
>
>
> > O-2: The paper's title mentions "recognition," yet the abstract refers to a broader usage of the benchmark, encompassing more than just classification (recognition).
>
> Thank you for your insightful comment. While the title of our paper emphasizes 'recognition', we provide bounding box information for each image in the dataset, facilitating its use for a wider range of tasks, such as object detection.
>
> However, the primary focus of our paper is on classification tasks. Even though the bounding box information expands the potential usability of our dataset to include object detection, this serves primarily as an auxiliary process to aid in the recognition of invasive species.
>
> > O-3: Four experimental settings are mentioned in abstract but only three are explicitly enlisted (supervised learning, semi-supervised learning, and self-supervised pretraining). Please clarify.
>
> Thank you for noting the discrepancy in the number of experimental settings mentioned. We acknowledge the oversight, and we appreciate your careful reading. The fourth experiment is indeed the CLIP zero-shot classification, which we detailed in the "Experiment on zero-shot classification" section in the Appendix.
>
> Due to space constraints, we did not include this in the main text. However, in light of your feedback, we will ensure that the revised version clearly mentions all four experimental settings in the abstract. We appreciate your attention to detail, which helps improve the clarity of our paper.

---

> ### Author Response · Authors · 2023-08-22
> **Review response for foZN, Part 2**
>
> ### Opportunities For Improvement:
> > O-4: Surprisingly, the masked image modeling is first mentioned in Section 4.4.2. A reader would not expect this part to be included in the study. Is it important? Why was it included?
>
> Thank you for your comment. Taking into account your feedback and my advisor's suggestions, we are indeed planning to rename section 4.2.2 as "Self-supervised Pretraining."
>
> The 'Masked image modeling' is a crucial aspect of our study and a topic of significant interest in the field of unsupervised learning in computer vision. It's incorporated into our study to compare the pre-training performance of different models on *Species-U* and *Imagenet-1K*. It serves as a vital tool for unsupervised pre-training on the *Species-U* dataset, and we will introduce this concept earlier in the revised version of the paper to improve the manuscript's flow.
>
> By including 'Masked image modeling' in our research, we hope our experimental findings will inspire more researchers to delve into related areas in *Species-196*.
>
> We have also incorporated 'Contrastive Learning' into our pretraining experiments, beyond the 'Masked Image Modeling' mentioned in our paper. In the revised version of the paper, we will include additional content on this topic. Notably, in our contrastive learning experiments, our Species196-U dataset again outperforms the Imagenet-1K under the same pretraining cost.
>
> Here is the supplementary contrastive learning experimental result table:
>
> | Models   | Method  | Batch Size | Pretrain Data | Pretrain Epoch | Linear_cls Epoch | Top-1 ACC |
> | :------- | :------ | :--------- | :------------ | :------------- | :--------------- | :-------- |
> | Resnet50 | swav    | 256        | In1k          | 200            | 100              | 69.0      |
> | Resnet50 | swav    | 256        | Species-U     | 200            | 100              | 69.7      |
> | Resnet50 | MoCo v2 | 256        | In1k          | 800            | 100              | 67.1      |
> | Resnet50 | MoCo v2 | 256        | Species-U     | 800            | 100              | 67.9      |
>
> > O-5: The paper requires thorough proofreading and correction of grammar issues, e.g., see reference [1] or “faces. we” (Section 3.2)
>
> Thank you for pointing out the grammatical issues in our paper. We assure you that we will conduct thorough proofreading and correct all grammar issues for the revised submission. We appreciate your attention to detail and your patience during this process.
>
> ### Limitations:
>
> > L-1: The study reports results of performance indices that may be affected by classes with a higher number of samples.
>
> Thank you for your observation and suggestion. We acknowledge that the Species196 dataset exhibits class imbalance, a characteristic common in many real-world datasets. Based on your suggestion (O-1), we will augment our performance evaluation in the revised version by including Macro F-1 scores. This will provide a more comprehensive measure of the models' performances and facilitate comparisons for researchers. We appreciate your valuable feedback.
>
>
>
> > L-2: As the labeling of bounding boxes for pest locations was carried out on randomly selected subsets of images, their quantity should be reported somewhere.
>
> Thank you for your query. We apologize for any confusion caused by the discrepancy between line 157-158 and 161-162 in our paper. To clarify, every image in the Species196-L dataset does indeed have bounding box annotations, contrary to what is suggested in lines 157-158.
>
> We acknowledge the error in our manuscript and appreciate your attention to detail. We will correct this in the revised version and make it clear that all 19236 images in the Species196-L dataset come with bounding box annotations.
>
> We hope this clarification addresses your concern, and we thank you again for your valuable feedback.
>
> > L-3: The download links at https://species-dataset.github.io/download.html are not permanent and appear to be more like personal (Google Drive) storage. Therefore, the availability and maintenance of the benchmark require serious attention.
>
> Thank you for your comment on the download links for the Species196 dataset. We acknowledge the importance of having stable and reliable links to ensure the continuous availability and maintenance of the benchmark.
>
> Currently, our dataset is hosted on Google Drive and Baidu Yunpan, and links are provided on our GitHub Pages website. We have chosen these platforms as they offer reliable service and are widely used, making them accessible to a broad range of researchers from different regions.
>
> We monitor the availability of the dataset and strive to promptly address any issues that arise. However, we understand your concern regarding the potential volatility of these links. We appreciate your feedback and will continue to seek ways to improve the **accessibility** and **persistence** of our dataset.

---

> > ### Comment · Reviewer_foZN · 2023-08-29
> >
> > Thank you for your answers and modifications. In my opinion, the revised manuscript is in a better shape of presentation. Consequently, I've updated the rating.

---

> > > ### Author Response · Authors · 2023-08-30
> > >
> > > Dear Reviewer,
> > >
> > > Thank you for the updated rating and your comments. It's glad to know that our additions, such as the macro F1 scores and the corrections based on your suggestions, have improved the quality of our paper. We deeply appreciate your guidance that greatly contributed to these improvements.
> > >
> > > Best regards,
> > > Author

---

### Official Review · Reviewer_3Z3U · 2023-07-21
**Review of Species-196**

**Rating:** 6
**Confidence:** 4
**Correctness:** The dataset construction and the expe…
**Clarity:** The presentation is very clear.

**Strengths:**

- The proposed dataset is very useful in practice. It contains a lot of expert annotations of invasive species at a larger scale than previous work. It will be useful for both practitioners who want to specifically tackle this problem, and computer vision researchers who are looking for a fine-grained classification benchmark.

- The final comparison and analysis is extensive with many approaches (supervised, semi-supervised, self-supervised), backbones (convnets and vision transformers) and tasks (detection and classification).

**Additional Feedback:**

The proposed dataset is a new contribution both useful for practical applications and interesting for researchers that tackle fine-grained classification problems. I am therefore leaning toward acceptance.

**Documentation:**

The data collection process is described in details, although I would appreciate more details and statistics on the provenance of the images.

**Ethics:**

No ethics concerns.

**Limitations:**

- Data imbalance which is a big limitation of the dataset is discussed.

**Opportunities For Improvement:**

- Could you give more details on the provenance of the images ? How many of them come from each dataset, how many of them are new.

- How do you ensure that the pictures are taken with a wide variety of poses and occlusion ? Do you think a model trained on your dataset would generalises to pictures taken in the wild with a phone or camera ?

- Could you give more specific details abut your comment L114-116 ? What does it mean to lower the difficulty in data acquisition ? Why do you think is has greater transfer potential ?

- It would be great to have an idea of the geographic repartition of the species present in your dataset, and more generally more info on the data so that it could be useful for practitioners at different locations on the planet.

- Do you check if roughly the data in Species196-U is in domain, and belongs to the 196 categories, or if there is a wide proportion of samples that belongs to other species, potentially none invasive ? With this information in mind, I would be curious to compare pretraining on Inat and pretraining on Species196-U for fine-tuning on Species196-L.

- How well do you do on rare classes ? Is it difficult to collect more data for these classes ? A deeper analysis would be interesting.

- The semi-supervised setup is unclear. Why do you not use the same backbone than your supervised experiment so that we can compare the approaches fairly ? The results are very weak compared to the supervised approach but it is hard to draw a conclusion.

- Have you tried to pretrain on both ImageNet and Species196-U and then fine-tune on Species196-L ?

**Relation To Prior Work:**

The author discussed in details the novelty of their dataset compared to existing datasets.

**Summary And Contributions:**

This paper proposes a new dataset of images of invasive species, and a fine-grained classification and detection benchmark. The dataset is divided into a small dataset of 19000 images with expert annotations, and a bigger dataset of 1.5M unlabelled images. The paper also proposes a comparison on the benchmark of different approaches, supervised, semi-supervised and self-supervised, using various backbones.

---

> ### Author Response · Authors · 2023-08-22
> **Review response for 3Z3U, Part 1**
>
> ### Opportunities For Improvement
>
> > O-1: Could you give more details on the provenance of the images ? How many of them come from each dataset, how many of them are new.
>
> Thank you for your question. All our data was obtained from publicly available sources on the internet, with the correctness of the data manually verified. We conducted an analysis based on the download URLs of the data, and the table below shows the top twenty sources for both *Species196-L*, which illustrates the diversity of our data sources. As you can see, the sum of the percentages from the top 20 sources is only 39.54%, indicating the richness and breadth of our dataset's sources. We owe a debt of gratitude to these open and reliable data sources, which have significantly contributed to the construction of the Species196 dataset.
>
> | NO.  | SPECIES-L (TOTAL 19236)                                      | COUNT | PROPORTION |
> | :--: | ------------------------------------------------------------ | :---: | :--------: |
> |  1   | [inaturalist-open-data.s3.amazonaws.com](https://www.inaturalist.org/) | 2457  |   12.77%   |
> |  2   | [static.inaturalist.org]()                                   | 1140  |   5.93%    |
> |  3   | [www.naturamediterraneo.com](www.naturamediterraneo.com)     |  612  |   3.18%    |
> |  4   | [thumbs.dreamstime.com]()                                    |  415  |   2.16%    |
> |  5   | [bugguide.net]()                                             |  391  |   2.03%    |
> |  6   | [v3.boldsystems.org]()                                       |  233  |   1.21%    |
> |  7   | [www.biolib.cz](www.biolib.cz)                               |  215  |   1.12%    |
> |  8   | [www.zin.ru](www.zin.ru)                                     |  209  |   1.09%    |
> |  9   | [www.researchgate.net](www.researchgate.net)                 |  193  |   1.00%    |
> |  10  | [i.servimg.com]()                                            |  191  |   0.99%    |
> |  11  | [www.plantarium.ru](www.plantarium.ru)                       |  190  |   0.99%    |
> |  12  | [www.lepiforum.de](www.lepiforum.de)                         |  185  |   0.96%    |
> |  13  | [inpn.mnhn.fr]()                                             |  178  |   0.93%    |
> |  14  | [www.galerie-insecte.org](www.galerie-insecte.org)           |  159  |   0.83%    |
> |  15  | [calphotos.berkeley.edu]()                                   |  148  |   0.77%    |
> |  16  | [c8.alamy.com]()                                             |  143  |   0.74%    |
> |  17  | [gd.eppo.int]()                                              |  142  |   0.74%    |
> |  18  | [www.discoverlife.org](www.discoverlife.org)                 |  139  |   0.72%    |
> |  19  | [www.biodiversidadvirtual.org](www.biodiversidadvirtual.org) |  138  |   0.72%    |
> |  20  | [upload.wikimedia.org]()                                     |  127  |   0.66%    |
> |      | Total                                                        |       |   39.54%   |
>
> > O-2: How do you ensure that the pictures are taken with a wide variety of poses and occlusion ? Do you think a model trained on your dataset would generalises to pictures taken in the wild with a phone or camera ?
>
> Thank you for your question. Our dataset's diversity, including varying poses and occlusion, is a natural result of the wide variety of internet sources we used. During collection, we selected images with diverse backgrounds. As can be seen in Appendix Figure 1, many images are from the wild. As a result, models trained on our dataset are expected to perform reasonably well with 'in-the-wild' photos taken by phones or cameras.
>
> > O-3：Could you give more specific details abut your comment L114-116 ? What does it mean to lower the difficulty in data acquisition ? Why do you think is has greater transfer potential ?
>
> Thank you for your question, and I apologize if our original statement was unclear. In many practical applications, the scale of available data often falls short of what datasets like iNat2017 and iNat2021 offer. It's also often challenging to gather user-contributed data through a large community, as is done with iNaturalist.
>
> In contrast, the scale and collection method of our Species196 dataset align more closely with real-world applications. Moreover, Species196 provides a challenging testbed for the research community to explore how to improve model's performance through limited meticulously labeled data and large-scale unlabeled data. This practicality and versatility is what we refer to when we speak of 'lowered difficulty in data acquisition' and 'greater transfer potential'.

---

> ### Author Response · Authors · 2023-08-22
> **Review response for 3Z3U, Part 2**
>
> ### Opportunities For Improvement
> > O-4: It would be great to have an idea of the geographic repartition of the species present in your dataset, and more generally more info on the data so that it could be useful for practitioners at different locations on the planet.
>
> Thank you for your feedback. Our Species196 dataset currently focuses on species listed in the *Catalogue of Quarantine Pests for Import Plants to China*. Considering that invasive species are a global concern and there is often many overlap of these species across different regions, we are exploring the possibility of identifying commonly occurring species across different regions in our future work. We appreciate your suggestion and are committed to advancing our research to potentially benefit the global community.
>
> > O-5: Do you check if roughly the data in Species196-U is in domain, and belongs to the 196 categories, or if there is a wide proportion of samples that belongs to other species, potentially none invasive ? With this information in mind, I would be curious to compare pretraining on Inat and pretraining on Species196-U for fine-tuning on Species196-L.
>
> Thank you for your question. A rough estimate would be about 90%. We've provided insights into this process in Appendix Figure 2 of our paper, where we displayed images retrieved at different similarity ranks. Even at rank 5000, the retrieved images still showed a high resemblance to the original species image.
>
> For the second part of your question, we conducted additional experiments under the same settings to compare the performance of Swin-B using SimMIM pretraining on both the iNat2021 dataset and our Species196-U dataset. We randomly sampled 1,200,000 images from the iNat2021 dataset for pretraining to ensure a fair comparison. The results are presented in the following table:
>
> | Model    | Method   | Dataset     | Pretraining Epoch | Finetuning Epoch | Accuracy (%) | Top-5 Accuracy (%) |
> | :------- | :------- | :---------- | :---------------- | :--------------- | :----------- | :----------------- |
> | Swin-B   | SimMIM   | ImageNet-1K | 800               | 100              | 80.5         | 94.5               |
> | Swin-B   | SimMIM   | Species-U   | 800               | 100              | **81.6**     | **95.0**           |
> | *Swin-B* | *SimMIM* | *Inat2021*  | *800*             | *100*            | 81.1         | 94.6               |
>
> Under the same parameters, Swin-B pretrained on iNat outperforms that pretrained on ImageNet-1K, but doesn't surpass our proposed
> *Species196-U*.
>
> > O-6: How well do you do on rare classes ? Is it difficult to collect more data for these classes ? A deeper analysis would be interesting.
>
> Thank you for your insightful question about the performance on rare classes and the challenges in data collection. For certain species, such as *Bruchidius incarnates* and *Schnaspis longirostris*, the available images for collection were indeed limited. These species aren't readily findable in biological databases like [iNaturalist ↗](https://www.inaturalist.org/) or [bugguide.net ↗](http://bugguide.net/), and the volume of accessible data on the internet is sparse.
>
> This situation leads to an imbalance in our data. However, we view this as a reflection of the real-world distribution and availability of species data. Some species are simply more obscure and less documented than others. This presents a unique opportunity to develop models robust enough to handle such real-world imbalances and still deliver accurate results. We're continually working to refine our model to better handle such situations.
>
> > O-7: The semi-supervised setup is unclear. Why do you not use the same backbone than your supervised experiment so that we can compare the approaches fairly ? The results are very weak compared to the supervised approach but it is hard to draw a conclusion.
>
> Thank you for your insightful question regarding the semi-supervised setup. We understand your concerns about the differences in the backbones used in the supervised and semi-supervised experiments, and the comparative performance of these approaches.
> 1. **Diverse Objectives**: In the supervised learning section, our objective was to establish benchmarks. In the semi-supervised and self-supervised sections, we aimed to illustrate different methodologies' application to our dataset, *Species-U*.
> 2. **Resource Constraints**: The original Noisy Student setup is computationally intensive. It requires generating pseudo-labels for a vast number of unlabeled images from large-scale datasets such as JFT and YFCC100M, and involves an iterative training process where the student model matures into the teacher for subsequent iterations. Our dataset, *Species-U*, with only 1.2 million images, required a more streamlined approach, leading us to opt for smaller models.
> In response to your feedback, we have conducted additional experiments. Supervised learning results of ResNet18 and ResNet34 will be included in Table 2.

---

> ### Author Response · Authors · 2023-08-22
> **Review response for 3Z3U, Part 3**
>
> ### Opportunities For Improvement
>
> > O-8: Have you tried to pretrain on both ImageNet and Species196-U and then fine-tune on Species196-L ?
>
> Thank you for your question. In response, we have conducted additional experiments and updated the table presented in response to Q5. We added a setting where pretraining was performed using both ImageNet and Species196-U. In this setting, we compared the results of fine-tuning 100 epochs after pretraining for both 400 and 800 epochs.
>
> | Model    | Method        | Dataset                   | Pretraining Epoch | Finetuning Epoch | Accuracy (%) | Top-5 Accuracy (%) |
> | :------- | :------------ | :------------------------ | :---------------- | :--------------- | :----------- | :----------------- |
> | Swin-B   | SimMIM    | ImageNet-1K               | 800               | 100              | 80.5         | 94.5               |
> | Swin-B   | SimMIM    | Species-U                 | 800               | 100              | 81.6         | 95.0               |
> | Swin-B   | SimMIM    | Inat2021                  | 800               | 100              | 81.1         | 94.6               |
> | *Swin-B* | *SimMIM* | *ImageNet-1K + Species-U* | *400*             | *100*            | *81.0*       | *94.8*             |
> | *Swin-B* | *SimMIM* | *ImageNet-1K + Species-U* | *800*             | *100*            | ***82.5***   | ***96.0***         |
>
> ### **Limitations:**
>
> > L-1: Data imbalance which is a big limitation of the dataset is discussed.
>
> Thank you for your comment on the data imbalance in our Species196 dataset. Indeed, there's a disparity in the number of images available for different species, reflecting the varied abundance and documentation of species in the wild.
>
> While this circumstance could be seen as a limitation, we prefer to view it as a characteristic of real-world biodiversity data. Such disparity presents a valuable opportunity for the development and evaluation of models that can handle class imbalance effectively, a common issue in numerous practical applications.
>
> To further aid in this research, we also offer the Species196-U dataset. This provides an additional avenue to explore how unsupervised learning can tackle real-world problems, such as optimizing performance when dealing with data imbalance and limited labeled data.

---

### Official Review · Reviewer_QxhT · 2023-08-03
**A Novel Fine-grained Dataset with Minor Issues**

**Rating:** 6
**Confidence:** 4

**Strengths:**

* Semi-supervised learning with open-world fine-grained images is a novel problem in the era of large datasets and large pre-trained models. This dataset covers a diverse and large set of 196 invasive species, larger and more diverse than most prior work. This is useful for developing practical invasive species recognition systems.

* This paper builds the labeled and unlabeled datasets with CLIP pre-trained models on LAION-5B, which is a novel strategy. The large unlabeled dataset Species196-U enables experiments on popular semi-supervised techniques, and baseline results take modern network architectures and self-supervised pre-training into consideration.

**Additional Feedback:**

No more questions.

**Clarity:**

The paper is well-written and easy to follow. The dataset collection, taxonomy information, and experiments are clearly explained. Providing more details on the unlabeled data collection would further improve clarity.

**Correctness:**

The claims and method seem technically sound, though more analysis of the labeled and unlabeled data would help better understand its challenges and usefulness. The benchmark experiments follow standard practices and are comprehensive.

**Documentation:**

The authors provided a download website but have not provided online documents that contain users’ guidance and benchmark results. I suggest the authors attach relevant material during the rebuttal. The documents of the online website should be completed for the benefit of the community.

**Limitations:**

* Compared to fine-grained classification datasets (e.g., CUB-200-2011 and iNaturalist variants), the proposed dataset is large enough, but the class number is relatively small. Meanwhile, the labeled 19K dataset is extremely imbalanced, and some classes might have limited data over 196 categories, which will be too difficult for semi-supervised learning settings.

* More semi-supervised learning baselines should be included in the benchmark results. Please refer to the semi-supervised learning benchmark USB [1] as the template. Popular semi-supervised methods like FixMatch [2] and semi-supervised fine-tuning experiments of pre-trained models [3] should be included.

Reference

[1] Yidong Wang, et al. USB: A Unified Semi-supervised Learning Benchmark for Classification. NeurIPS, 2022.

[2] Kihyuk Sohn, et al. FixMatch: Simplifying Semi-Supervised Learning with Consistency and Confidence. NeurIPS, 2020.

[3] Ting Chen, et al. Big Self-Supervised Models are Strong Semi-Supervised Learners. NeurIPS, 2020.

**Opportunities For Improvement:**

* Details on criteria for unlabeled image retrieval could be expanded, which would help establish relevance and reproduction for other researchers. Meanwhile, more analysis could be done on diversity within each class between the labeled and unlabeled data to evaluate the quality and reliability of the built datasets.

* Refer to the limitations for more improvement suggestions.

**Relation To Prior Work:**

Related datasets are adequately compared and discussed, e.g., differences like taxonomy information are highlighted. However, the semi-supervised learning literature should be expanded according to USB [1].

**Summary And Contributions:**

This paper introduces Species196, a new fine-grained image classification dataset with 196 invasive species. Concretely, the dataset contains 19K manually labeled images across different classes of invasive species like insects, mollusks, and weeds (Species196-L), while containing a larger set of 1.2M unlabeled images crawled from the web (Species196-U) based on LAION-5B. In Species196-U, each image is queried by performing an image-to-image search in LAION-5B dataset with CLIP pretrained ViT-L/14. The paper provides taxonomy information, and benchmark results using state-of-the-art CNNs and Transformers in semi-supervised learning settings.

---

> ### Author Response · Authors · 2023-08-22
> **Review response for QxhT**
>
> ### Opportunities For Improvement
>
> > O-1: Details on criteria for unlabeled image retrieval could be expanded, which would help establish relevance and reproduction for other researchers. Meanwhile, more analysis could be done on diversity within each class between the labeled and unlabeled data to evaluate the quality and reliability of the built datasets.
>
> Thank you for your comments and suggestions. For the *Species196-U* dataset, as detailed in the manuscript on lines 171-180, page 6, we randomly sampled three images from each category and retrieved 8,000 unlabeled images per class. Post duplicate images removal, our final dataset consisted of roughly 1.5 million unique images. To establish a fair comparison with ImageNet-1K pretraining, we selected 1.2m image-text pairs from the collected 1.5 million data points.
>
> We did not categorize each species precisely in the *Species196-U* dataset due to the inter-class similarities leading to duplicate image retrievals across different categories. However, we provide super-classes for  *Species196-U* dataset, such as insects, molluscs, weeds, and snails.
>
> Since there is  a 1% download link rot per month, we've provided the download URLs with 1.5 million pics. Users are encouraged to randomly sample 1.2 million images to validate our experimental results.
>
> ### Limitations:
>
> > L-1: Compared to fine-grained classification datasets (e.g., CUB-200-2011 and iNaturalist variants), the proposed dataset is large enough, but the class number is relatively small. Meanwhile, the labeled 19K dataset is extremely imbalanced, and some classes might have limited data over 196 categories, which will be too difficult for semi-supervised learning settings.
>
> Thank you for your question. The Species196 dataset was designed to address key challenges in regional invasion biometric recognition. During the data collection process, we aimed to maintain a sample size ranging between 50-200 for each category. However, for certain classes such as *IBruchidius incarnates* and *Schnaspis longirostris*, the number of available images was limited. This imbalance, we believe, mirrors the realities of the natural world where some categories are inherently more abundant or easier to capture than others. Rather than considering this as a limitation, we see it as a challenge for our dataset.
>
> We understand that compared to fine-grained classification datasets like CUB-200-2011 and iNaturalist variants, our dataset has fewer classes. However, it's the real-world relevance and challenge that our dataset presents which we believe will be of interest and value to the research community. We look forward to seeing how researchers tackle these challenges to advance the state of the art in machine learning.
>
> > L-2: More semi-supervised learning baselines should be included in the benchmark results. Please refer to the semi-supervised learning benchmark USB [1] as the template. Popular semi-supervised methods like FixMatch [2] and semi-supervised fine-tuning experiments of pre-trained models [3] should be included.
>
> Thank you for your insightful suggestions. As per your advice, we have expanded our experiments to include additional popular semi-supervised methods like Mixmatch, FixMatch and RemixMatch. In these experiments, we treated  the *Species196-L* as labeled data and *Species196-U* as unlabeled data,. The results of these experiments using Resnet50 are as follows:
>
> | Models   | Method     | Top1 ACC(only Species-L) | Top1 ACC(semi-supervised using Specis-U) |
> | :------- | :--------- | :----------------------: | :--------------------------------------: |
> | Resnet50 | Mixmatch   |          64.32           |                  67.87                   |
> | Resnet50 | Fixmatch   |          64.32           |                  66.94                   |
> | Resnet50 | Remixmatch |          64.32           |                  69.30                   |
>
> In addition, we have also conducted experiments using contrastive learning methods similar to SimCLR such as Swav and MoCo v2. The linear probe results of *Species196-L* are as follows:
>
> | Models   | Method  | Batch Size | Pretrain Data | Pretrain Epoch | Linear_cls Epoch | Top-1 ACC |
> | :------ | :----- | :-------- | :----------- | :------------ | :-------------- | :------- |
> | Resnet50 | swav    | 256        | In1k          | 200            | 100              | 69.0      |
> | Resnet50 | swav    | 256        | Species-U     | 200            | 100              | 69.7      |
> | Resnet50 | MoCo v2 | 256        | In1k          | 800            | 100              | 67.1      |
> | Resnet50 | MoCo v2 | 256        | Species-U     | 800            | 100              | 67.9      |
>
> Our results indicate that *Species-U* can contribute effectively to the performance of models on *Species-L* using the methods you proposed. Our team is committed to conducting further research and integrating additional methodologies to validate the efficacy of our *Species196-U* dataset.

---

> > ### Comment · Reviewer_QxhT · 2023-08-29
> >
> > I thank the authors for the detailed responses to my concerns, and the problems have been settled. I hope the author to further polish the writing and the dataset webpage (e.g., online documents, etc.) and also wish this dataset facilitates the community of semi-supervised learning and fine-grained classification.

---

> > > ### Author Response · Authors · 2023-08-30
> > >
> > > Dear Reviewer,
> > >
> > > We genuinely appreciate your feedback and we're glad that your concerns have been addressed. Your suggestion on enhancing our writing and the Species196 dataset webpage are indeed valuable. We will continue to refine our *Species196* dataset and the accompanying documentation to contribute positively to the research community.
> > >
> > > Best regards,
> > > Author

---

### Comment · Area_Chair_Eiae · 2023-08-29
**Reviewers please respond to authors**

Dear Reviewers,

We are approaching to the end of the author-reviewer discussion period. We kindly request that you review the authors' rebuttal (if you have not) and indicate whether your concerns have been satisfactorily addressed. If your concerns are well addressed, please also update your scores accordingly.

Thank you for your valuable contributions to this process.

---

### Decision · Program_Chairs · 2023-09-22

**Decision:**

Accept (Poster)

**Comment:**

This paper received overall positive reviews. Reviewers generally regard the problem and the way of data construction novel, the dataset useful for fine-grained classification study, and the comparison and analysis extensive. They also raised some concerns and suggestions: clarification on some algorithm and dataset construction details, the data imbalance problem, evaluation with more and sophisticated algorithms, writing quality, etc. The rebuttal was persuasive and addressed most concerns. The final ratings unanimously recommend acceptance. The AC checked the paper, rebuttal, and review comments, and recommends accepting the paper.